# Kill Two Birds with One Stone: Rethinking Data Augmentation for Deep Long-tailed Learning

**Binwu Wang[2], Pengkun Wang[2,3\*], Wei Xu[2], Xu Wang[2,3], Yudong Zhang[2],**
**Kun Wang[2], Yang Wang[1,2,3\*]**
[1]Key Laboratory of Precision and Intelligent Chemistry, University of Science and Technology of China, Hefei, China
[2]University of Science and Technology of China (USTC), Hefei, China
[3]Suzhou Institute for Advanced Research, USTC, Suzhou, China
`wbw1995@mail.ustc.edu.cn, pengkun@ustc.edu.cn`
`weixu1223@mail.ustc.edu.cn, wx309@ustc.edu.cn`
`{zyd2020,wk520529}@mail.ustc.edu.cn, angyan@ustc.edu.cn`

## Abstract

Real-world tasks are universally associated with training samples that exhibit a long-tailed class distribution, and traditional deep learning models are not suitable for fitting this distribution, thus resulting in a biased trained model. To surmount this dilemma, massive deep long-tailed learning studies have been proposed to achieve inter-class fairness models by designing sophisticated sampling strategies or improving existing model structures and loss functions. Habitually, these studies tend to apply data augmentation strategies to improve the generalization performance of their models. However, this augmentation strategy applied to balanced distributions may not be the best option for long-tailed distributions. For a profound understanding of data augmentation, we first theoretically analyze the gains of traditional augmentation strategies in long-tailed learning, and observe that augmentation methods cause the long-tailed distribution to be imbalanced again, resulting in an intertwined imbalance: *inherent data-wise imbalance* and *extrinsic augmentation-wise imbalance*, i.e., two 'birds' co-exist in long-tailed learning. Motivated by this observation, we propose an adaptive Dynamic Optional Data Augmentation (DODA) to address this intertwined imbalance, i.e., one 'stone' simultaneously 'kills' two 'birds', which allows each class to choose appropriate augmentation methods by maintaining a corresponding augmentation probability distribution for each class during training. Extensive experiments across mainstream long-tailed recognition benchmarks (e.g., CIFAR-100-LT, ImageNet-LT, and iNaturalist 2018) prove the effectiveness and flexibility of the DODA in overcoming the intertwined imbalance.

## 1 Introduction

With the maturity of deep learning LeCun et al. (2015), massive deep models show extraordinary performance on large-scale curated datasets (e.g., ImageNet Russakovsky et al. (2015) and CIFAR-100 Cao et al. (2019)). However, balanced artificial datasets do not conform to the data distribution (e.g., class imbalance) of real-world applications. Once facing imbalanced datasets, the performance of the deep models trained by the common practice of empirical risk minimization Vapnik (1991) will decrease significantly, e.g., the model can be easily biased towards majority classes.

Recently, massive deep long-tailed learning studies have been proposed to surmount the class imbalance problem. The most intuitive and mainstream paradigm is class re-balancing, which balances the training sample numbers or weights of different classes during model training by re-sampling Kang et al. (2020); Ren et al. (2020); Wang et al. (2020); Jia et al. (2023) or cost-sensitive

---

*Prof. Yang Wang and Pengkun Wang are the corresponding authors.

learning Lin et al. (2017); Cui et al. (2019); Tan et al. (2020); Guo et al. (2022); Guo & Li (2022). Although the average performance is improved, class rebalancing methods cannot essentially handle the issue of lacking information, particularly on tail classes due to limited data amount Zhang et al. (2021b). To break through this restraint, some studies seek to transfer the knowledge from head classes to enhance model training on tail classes Yin et al. (2019); Kim et al. (2020b).

Another flexible line of research is to directly apply data augmentation methods to enhance the quantity and quality of training samples from the perspectives of data and representation. For example, FASA Zang et al. (2021) proposed to generate class-wise representations based on a Gaussian prior to augment the under-represented tail classes. Remix Chou et al. (2020) and VideoLT Zhang et al. (2021a) introduced a re-balanced mixup method to particularly enhance tail classes. However, simply using existing class-independent augmentation strategies for improving long-tailed learning is unfavorable, since they may further increase imbalance considering head classes have more samples and would be augmented more Zhang et al. (2021b). Considering this, as a pioneer, FSR Wang et al. (2023) first proposed an adaptive augmentation to rebalance the potential temporal feature space from the data perspective. CUDA Ahn et al. (2023) further proposed a simple algorithm to find the proper class-wise augmentation strength through curriculum learning Hacohen & Weinshall (2019). Despite extensive recent studies, an unresolved problem is *whether it is optimal to utilize the same kind of augmentation strategy on all classes for long-tailed learning.*

With this question, we first theoretically analyze the gains of traditional augmentation strategies in long-tailed learning. Inspired by Balestriero et al. (2022), as shown in Figure 1, we explain that data augmentation (DA) is not always beneficial to long-tailed learning, i.e., *DA will potentially sacrifice certain classes (especially tail classes) while improving the model performance*. This means that two 'birds' co-exist in long-tailed learning: inherent data-wise imbalance and extrinsic augmentation-wise imbalance. The former is the inherent property of real-world data distributions, while the latter is the side effect caused by DA when improving model generalization. We also conduct extensive experiments on CIFAR-100-LT with imbalance ratios (IR) of 50 and 100

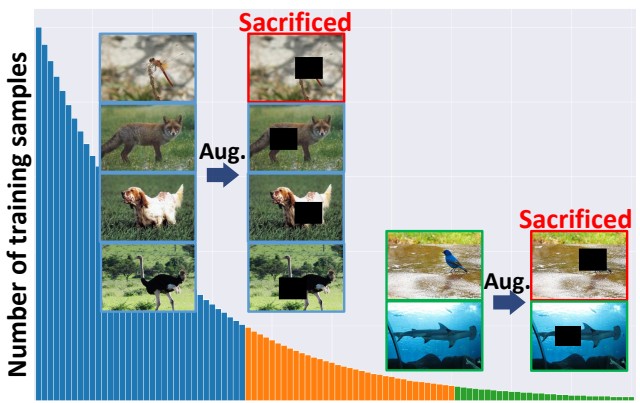

Figure 1: Motivation of DODA. In deep long-tailed learning, traditional data augmentation can significantly improve the average performance of the model, but it will potentially sacrifice certain classes (i.e., red box), especially tail classes.

to confirm the explanation. From the results shown in Figure 2, we can observe that DA caused a significant reduction in performance for some classes (e.g., most tail classes), which *runs counter to the purpose of long-tailed learning*.

To this end, we propose an adaptive 'stone' called Dynamic Optional Data Augmentation (DODA) to 'kill' this intertwined imbalance, which allows each class to choose appropriate augmentation methods during training. Specifically, to avoid the sacrifice caused by class-independent DA, we maintain a '*preference list*' for each class (i.e., a probability distribution of DAs being selected) during training, which is the basis for each class to choose applied augmentation methods. Then, to make this 'list' more precise, we adjust the corresponding probability distribution according to the positive sample size of each class. In this way, this 'list' will be dynamically corrected during training, thus each class will choose the most beneficial augmentation method to avoid being sacrificed, thereby reducing the overall sacrifice probability. We integrate DODA with various long-tailed learning methods and prove that DODA can significantly improve the model performance and have high flexibility. Furthermore, we conduct several experiments to compare DODA with existing DAs for long-tailed learning and show that DODA achieves state-of-the-art performance.

Our contributions in this paper are summarized as follows:

- *New problem and insight*: for the first time, we theoretically analyze the gains of traditional augmentation strategies in long-tailed learning: DA will potentially sacrifice certain classes (especially tail classes) while improving the model performance, thus we need to discreetly handle the inherent data-wise imbalance and extrinsic augmentation-wise imbalance.

- *New advisable augmentation*: to 'kill' these two 'birds', we propose an adaptive 'stone' called Dynamic Optional Data Augmentation (DODA) to allow each class to choose appropriate augmentation methods during training.

- *Compelling empirical results*: DODA achieves the state-of-the-art performance across mainstream long-tailed benchmarks including CIFAR-100-LT, ImageNet-LT, and iNaturalist 2018.

## 2 RETHINKING DA: IS DA ALWAYS BENEFICIAL TO LONG-TAILED LEARNING?

Applying DA in long-tailed learning has been empirically proven to significantly improve the average accuracy of the deep model. This overall improvement is encouraging, but it also prompts us to think about *how DA brings gains to the problem*. In this section, we will analyze this further.

### 2.1 MOTIVATION: DA IN LONG-TAILED LEARNING IS 'HYPOCRITICAL'

**Preliminary.** Class-independent DA is the mainstream strategy in long-tailed learning, which is simple but effective (i.e., average improvement). However, recent work has found that DA can result in unfair model complexity control across different classes, leading to the deep model that achieve an overall accuracy improvement but perform poorly on some classes. This phenomenon reflects the fact that DA actually produces a *hypocritical* improvement. In a class-balanced setting, Balestriero et al. (2022) explains this from the perspective of level-set. Formally, given a training dataset $\mathcal{D} = \{(x_i, y_i)|y_i \triangleq f^*(x_i)\}_{i=1}^N$, our goal is to learn a deep model $f_\theta$ to approximate the ideal model $f^*$ as closely as possible. To improve the generalization performance of the model, a DA strategy $\mathcal{O}(\cdot)$ will be habitually applied to $x$, which is a combination of one or more augmentation methods. During this approximation, the model that we intuitively consider to be ideal may already be biased.

**Theorem 1.** *Augmented samples produced by $\mathcal{O}(\cdot)$ do not respect the level-set of $f^*$. When we approximate the ideal model $f^*$ by minimizing the training loss (i.e., 0 training error), the latter tends to zero, while the former is greater than zero due to the augmented samples deviate from the level set of $f^*$. In this case, the trained model $f_\theta$ is biased compared to the ideal model $f^*$.*

$$\sum_{(x,y)\in\mathcal{D}} \mathbb{E}[||y - f^*(\mathcal{O}(x))||_2^2] > 0 \land \sum_{(x,y)\in\mathcal{D}} \mathbb{E}[||y - f_\theta(\mathcal{O}(x))||_2^2] = 0 \implies bias \quad (1)$$

Furthermore, we rethink the problem from the perspective of long-tailed learning. For each class in the long-tailed distribution dataset, such as class $c$, we also hope that the trained model $f_\theta$ can successfully predict the labels of all samples of this class. Therefore, $f_\theta$ also has the above bias in the class dimension. This explanation is formalized by the following theorem.

**Theorem 2.** *Under the long-tailed distribution, minimizing the training loss (i.e., 0 training error) is equivalent to minimizing the training loss for each class. For class $c$ and augmented samples $\{(\mathcal{O}(x), y)|y = c, (\mathcal{O}(x), y) \in \mathcal{D}\}$, the ideal trained model can minimize the training loss of class $c$, but when $\mathcal{O}(x)$ deviates from the level-set of $f^*$, DA will cause irreducible class-wise bias in $f_\theta$.*

$$\sum_{(x,y)\in\mathcal{D}|y=c} \mathbb{E}[||y - f^*(\mathcal{O}(x))||_2^2] > 0 \land \sum_{(x,y)\in\mathcal{D}|y=c} \mathbb{E}[||y - f_\theta(\mathcal{O}(x))||_2^2] = 0 \implies \text{class-wise bias}$$

$$(2)$$

Theorem 2 states that for long-tailed distributions, a DA cannot guarantee that it is label-preserving on all classes, which leads to some classes being sacrificed (i.e., seriously deviating from the level-set) after DA, resulting in irreducible class-wise bias. The detailed proof is in **Appendix A**. Based on this theorem, we observed the class-wise bias caused by various DAs under two imbalance rate settings (IR = {50, 100}) of CIFAR-100-LT, including simple augmentation (Cutout), flexible augmentation

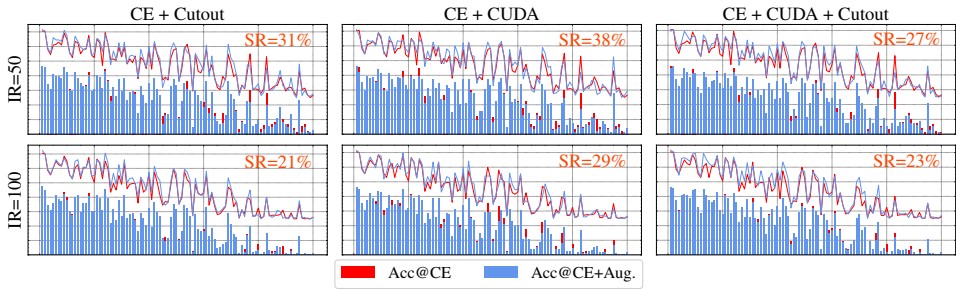

Figure 2: Impact of class-independent long-tailed DAs on classification accuracy on CIFAR-100-LT.

(CUDA), and hybrid augmentation (Cutout with CUDA). From the experimental results shown in Figure 2, we analyze the sacrifice rate (SR) under each setting and make the following findings: (1) All three types of DA can improve the average model performance (i.e., average accuracy); (2) The improvement in average performance inevitably sacrifices some classes, whether they are head or tail classes; (3) The phenomenon of sacrificing classes is especially evident on tail classes.

The experimental results are consistent with the statement of Theorem 2, which proves that the traditional class-independent DA is 'hypocritical' and will potentially sacrifice some classes (especially tail classes) when improving the average performance of the long-tailed learning model, which runs counter to the purpose of long-tailed learning.

## 2.2 DA FAVORS LONG-TAILED LEARNING THROUGH 'BULLYING'

As mentioned in Section 2.1, class-independent DA is hypocritical, and it shows its effectiveness by pleasing the 'strong', while tail classes are more likely to be regarded as the bullied 'weak'. To explain this phenomenon, we analyze the data distribution shift of different classes before and after DA from the perspective of feature space. In high-dimensional space, samples of the same class are usually closer to each other and form approximate clusters due to their inherent similarities. When applying the same data augmentation, the features in high-dimensional space undergo shifts that are consistent across different classes. To intuitively understand the impact of data augmentation on features of different classes, we approximate the high-dimensional space as a regular hyperspace and use the variation in the distribution span to represent the diversity improvement and high-dimensional space expansion caused by data augmentation. Intuitively, we approximate the feature space of classes to two-dimensional space to illustrate this. And we also provide a theoretical explanation in high-dimensional feature space in **Appendix A**.

**Definition 1** (Distribution Span). *Approximating the distribution of each class in the training set as a circle in a two-dimensional feature space, the distribution center of class $c$ is defined as $(\mathbb{X}_c, \mathbb{Y}_c)$ and the distribution radius is $\mathbb{R}_c$. The distribution span $\mathbb{S}_c$ can be expressed as follows:*

$$\mathbb{S}_c \Rightarrow (\mathbb{X} - \mathbb{X}_c)^2 - (\mathbb{Y} - \mathbb{Y}_c)^2 = \mathbb{R}_c^{\ 2} \tag{3}$$

For intuitive understanding, we assume that the data distribution of head class $c_h$ and tail class $c_t$ are $\mathbb{S}_{c_h}$ and $\mathbb{S}_{c_t}$, respectively. We uniformly apply the same DA $\mathcal{O}(\cdot)$ to the whole training set. Furthermore, the distribution span after DA can be defined as follows:

$$\bar{\mathbb{S}}_{c_h} \Rightarrow (\mathbb{X} - \mathbb{X}_{c_h})^2 - (\mathbb{Y} - \mathbb{Y}_{c_h})^2 = (\mathbb{R}_{c_h} + \Delta_{c_h})^2 \tag{4}$$

$$\bar{\mathbb{S}}_{c_t} \Rightarrow (\mathbb{X} - \mathbb{X}_{c_t})^2 - (\mathbb{Y} - \mathbb{Y}_{c_t})^2 = (\mathbb{R}_{c_t} + \Delta_{c_t})^2 \tag{5}$$

Here, $\Delta_{c_h}$ and $\Delta_{c_t}$ represent the increase in distribution radius within each class after DA. For the same augmentation method, $\Delta_{c_h} = \Delta_{c_t}$. Based on the increase in distribution radius, we analyze the sensitivity of each class to this augmentation.

**Theorem 3.** *The augmented data distribution is a combination of the original data distribution (base space) and the expanded data distribution (marginal space). The augmentation sensitivity $\psi$ can be defined as the ratio of the marginal space to the base space. Under the same DA, tail classes have a higher augmentation sensitivity, indicating that tail classes are more sensitive to the marginal space.*

$$\psi_{c_t} - \psi_{c_h} = 2\Delta_{c_h} \cdot \frac{\mathbb{R}_{c_h} - \mathbb{R}_{c_t}}{\mathbb{R}_{c_h}\mathbb{R}_{c_t}} + \Delta_{c_h}^{\ 2} \cdot \frac{\mathbb{R}_{c_h}^{\ 2} - \mathbb{R}_{c_t}^{\ 2}}{\mathbb{R}_{c_h}^{\ 2}\mathbb{R}_{c_t}^{\ 2}} > 0 \tag{6}$$

The main idea of the proof, provided in **Appendix A**, is to demonstrate that tail classes are more likely to have label non-preservation compared to head classes after DA, resulting in a greater deviation between the level-set learned by $f_\theta$ and the level-set of $f^*$ on tail classes.

In summary, DA in long-tailed learning is hypocritical and bullying, as it potentially sacrifices certain classes, especially tail classes, while boosting the average performance. Therefore, we not only need to deal with the *inherent data-wise imbalance* caused by the traditional sample distribution but also pay attention to the side effects of DA, the *extrinsic augmentation-wise imbalance*. It is frustrating that the existing 'stones' cannot kill both 'birds', so we need to *consider how to achieve a simple and effective DA to address this intertwined imbalance*.

## 3 DYNAMIC OPTIONAL DATA AUGMENTATION: AN ADVISABLE STRATEGY

In this section, based on the aforementioned theoretical analysis, we propose an advisable 'stone' called Dynamic Optional Data Augmentation (DODA) to address the intertwined imbalance. *The core philosophy of DODA is to allow each class to choose appropriate augmentation methods during training by maintaining a 'preference list' for each class.* The detailed algorithm overview of DODA is shown in Figure 3.

### 3.1 CLASS-WISE PREFERENCE LIST CONSTRUCTION

Firstly, we assume the existence of $K$ optional DAs and define the index of each DA as $k \in \{1, 2, ..., K\}$. These DAs are task-specific, such as Gaussian blur, rotation, and horizontal flip. Each DA $\mathcal{O}^k(\cdot) : \mathbb{R}^d \to \mathbb{R}^d$ has its predefined augmentation function and strength. For traditional strategies, we pre-determine the selected DAs and use them for all classes during training. However, this class-independent DA may sacrifice certain classes. Therefore, we maintain an augmentation preference

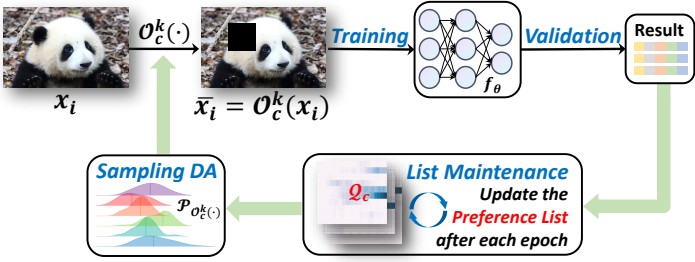

Figure 3: Overview of DODA. DODA allows each class to choose appropriate augmentation methods during training by maintaining a 'preference list' for each class.

list $\mathcal{Q}_c \ \forall_c \in \{1, ..., C\}$ for each class during training. This list records the *optional DAs* for each class $c$ and the *probability of each DA being selected*.

**Definition 2** (Probability of Each DA Being Selected). *The augmentation preference list $\mathcal{Q}_c$ for class $c$ records the selection hierarchy $\mathcal{Q}_c^k \ \forall_k \in \{1, ..., K\}$ of the $K$ optional DAs. Based on the selection hierarchy, we define the probability of each DA $\mathcal{O}^k(\cdot)$ being selected as follows:*

$$\mathcal{P}_{\mathcal{O}_c^k(\cdot)} = \frac{|\mathcal{Q}_c^k|}{\sum_{j \in \mathcal{Q}_c} |j|}, \quad \text{where } \mathcal{Q}_c = \{\mathcal{Q}_c^1, \mathcal{Q}_c^2, ..., \mathcal{Q}_c^K\} \tag{7}$$

Based on this preference list, we perform DA on the original dataset $\mathcal{D}$ and define the augmented dataset as $\bar{\mathcal{D}} = \{(\bar{x}_i, y_i) | (x_i, y_i) \in \mathcal{D}\}$. To preserve the knowledge of the original dataset $\mathcal{D}$, we define the augmentation probability as $p_{aug} < 1$, so each sample has a probability of $p_{aug}$ of being augmented. When performing DA, we randomly decide whether to augment the current sample with $p_{aug}$. Therefore, the augmented sample can be represented as follows:

$$\bar{x}_i = \begin{cases} \mathcal{O}_c^k(x_i), & \text{with prob. } p_{aug} \\ x_i, & \text{otherwise} \end{cases} \tag{8}$$

### 3.2 CLASS-WISE PREFERENCE LIST MAINTENANCE

On the augmented dataset $\bar{\mathcal{D}}$, we apply a long-tailed learning algorithm $\mathcal{F}$ to learn the desired deep model $f_\theta$, i.e., $\mathcal{F}(f_\theta, \bar{\mathcal{D}})$. It is worth mentioning that the choice of $\mathcal{F}$ is flexible, for example, we test various long-tailed learning algorithms in subsequent experiments.

During the early stages of training, the augmentation preference list set for one class may be inaccurate, meaning that there is still a high probability to select DAs that are detrimental to this class. Therefore, continually updating this list during training is necessary. *The core philosophy of updating is to up-weight the positive DAs and down-weight the negative DAs.* After each training epoch, we count the number of correctly predicted samples in each class, i.e., the positive sample size for each class. Formally, for class $c$, its positive sample size is defined as follows:

$$\nabla_{z_c^k}^{pos} = \sum_{(\bar{x}_i, y_i) \in \bar{\mathcal{D}} | y_i = c} \mathbb{1}_{\{f_\theta(\bar{x}_i) = c\}} \tag{9}$$

**Definition 3** (Augmentation Dominance). *For the dataset $\mathcal{D}$ and class $c$, DA $\mathcal{O}^{k_1}$ is said to dominate DA $\mathcal{O}^{k_2}$ on class $c$ if and only if $\nabla_{z_c^{k_1}}^{pos} > \nabla_{z_c^{k_2}}^{pos}$, where $\nabla_{z_c^{k_1}}^{pos} = \sum_{(x_i, y_i) \in \mathcal{D} | y_i = c} \mathbb{1}_{\{f_\theta(\mathcal{O}^{k_1}(x_i)) = c\}}$ and $\nabla_{z_c^{k_2}}^{pos} = \sum_{(x_i, y_i) \in \mathcal{D} | y_i = c} \mathbb{1}_{\{f_\theta(\mathcal{O}^{k_2}(x_i)) = c\}}$.*

**Theorem 4.** *The level-set bias $\delta(Q, P)$ can be defined as the degree of distributional deviation between the level-sets $Q$ and $P$. Suppose the level-set of the model $f_\theta$ trained on the original dataset is $P$, and the level-sets of the models $f_\theta^{k_1}$ and $f_\theta^{k_2}$ learned using DA $\mathcal{O}^{k_1}$ and DA $\mathcal{O}^{k_2}$, respectively, are $Q^{k_1}$ and $Q^{k_2}$. If the augmentation dominance of $\mathcal{O}^{k_1}$ is higher than that of $\mathcal{O}^{k_2}$, then the bias of $Q^{k_1}$ from $P$ is smaller than that of $Q^{k_2}$ from $P$, i.e., $\delta(Q^{k_1}, P) < \delta(Q^{k_2}, P)$.*

Theorem 4 states that a more dominant DA tends to avoid severe distribution bias. The detailed proof is in **Appendix A**. Therefore, to explore optional DA, we record the positive sample size $temp_c$ from the previous epoch for class $c$ to facilitate observation of whether the DA used in this epoch brings benefits to the class.

Based on the positive sample sizes from the current epoch and the previous epoch, we update the augmentation preference list for each class. For class $c$ and DA $\mathcal{O}_c^k$ used in the current epoch, if the positive sample size from the current epoch is greater than that from the previous epoch, we consider this DA is beneficial for class $c$, and thus we up-weight this DA that has a positive impact. Conversely, if this DA has a negative impact on class $c$, we down-weight it.

By dynamically maintaining the augmentation preference lists, DODA achieves adaptive class-dependent augmentation, which is not 'hypocritical' and not 'bullying', allowing each class (especially the tail classes) to choose appropriate augmentation methods and avoid being potentially sacrificed during training. To better illustrate the execution process of DODA, we provide a detailed execution flow in Algorithm 1. The code is available in `https://github.com/pongkun/Code-for-DODA`.

---

**Algorithm 1: DODA**

**Input:** Dataset $\mathcal{D} = (x_i, y_i)_{i=1}^N$, algorithm $\mathcal{F}$, training epoch $E$, number of optional aug. $K$, aug. probability $p_{aug}$.
**Output:** trained model $f_\theta$.
**Initialize:** Weight of each DA for each class $\mathcal{Q}_c^k = 1 \ \forall_k \in \{1, ..., K\}, \forall_c \in \{1, ..., C\}$.
**for** $e \le E$ **do**
 **for** $c \le C$ **do**
  Randomly select a DA $\mathcal{O}_c^k(\cdot)$ for class $c$ according to the weight distribution $\mathcal{Q}_c$.

$$\mathcal{P}_{\mathcal{O}_c^k(\cdot)} = \frac{|\mathcal{Q}_c^k|}{\sum_{j \in \mathcal{Q}_c} |j|}$$

  where $\mathcal{Q}_c = \{\mathcal{Q}_c^1, \mathcal{Q}_c^2, ..., \mathcal{Q}_c^K\}$
 **end**
 Generate $\bar{\mathcal{D}} = \{(\bar{x}_i, y_i) | (x_i, y_i) \in \mathcal{D}\}$ where

$$\bar{x}_i = \begin{cases} \mathcal{O}_c^k(x_i), & \text{with prob. } p_{aug} \\ x_i, & \text{otherwise.} \end{cases}$$

 Run LTL algorithm $\mathcal{F}$ using $\bar{\mathcal{D}}$, i.e., $\mathcal{F}(f_\theta, \bar{\mathcal{D}})$.
 **for** $c \le C$ **do**
  Compute positive sample size $\nabla_{z_c^k}^{pos}$ for class $c$

$$\nabla_{z_c^k}^{pos} = \sum_{(\bar{x}_i, y_i) \in \bar{\mathcal{D}} | y_i = c} \mathbb{1}_{\{f_\theta(\bar{x}_i) = c\}}$$

  **if** $\nabla_{z_c^k}^{pos} > temp_c$ **then** $\mathcal{Q}_c^k \longleftarrow \mathcal{Q}_c^k + 1$
   // Up-weight the positive DA
  **elif** $\nabla_{z_c^k}^{pos} = temp_c$ **then** $\mathcal{Q}_c^k \longleftarrow \mathcal{Q}_c^k$
  **else**  // Down-weight the negative DA
   **if** $\mathcal{Q}_c^k > 1$ **then** $\mathcal{Q}_c^k \longleftarrow \mathcal{Q}_c^k - 1$
   **else** $\mathcal{Q}_c^k \longleftarrow 1$
  $temp_c = \nabla_{z_c^k}^{pos}$
 **end**
**end**

---

## 4   EXPERIMENTS

In this section, we conduct empirical evaluations on multiple mainstream datasets to demonstrate the superiority of the proposed DODA in long-tailed learning.

### 4.1   EXPERIMENTAL SETTINGS

**Datasets.** To ensure a fair comparison, we conducted experiments on three mainstream long-tailed image recognition benchmarks including CIFAR-100-LT Cao et al. (2019), ImageNet-LT Liu et al.

Table 1: Accuracy (%) on CIFAR-100-LT dataset (Imbalance ratio={10, 50, 100}) wtih state-of-the-art methods. **Blod** indicates the best performance while underline indicates the second best. (+) and (-) indicate the the relative gain. We report the average results of three random trials.

| Method | IR=10 | | | | IR=50 | | | | IR=100 | | | |
|---|---|---|---|---|---|---|---|---|---|---|---|---|
| | Head | Medium | Tail | All | Head | Medium | Tail | All | Head | Medium | Tail | All |
| CE He et al. (2016) | 63.2 | 40.3 | - | 56.5 (+0.0) | 63.9 | 36.2 | 15.2 | 43.8 (+0.0) | 65.6 | 36.2 | 8.2 | 38.1 (+0.0) |
| CE + CMO Park et al. (2022) | 67.0 | 45.0 | - | 60.2 (+3.7) | 68.6 | 37.8 | 18.7 | 47.0 (+3.2) | 70.1 | 40.6 | 10.3 | 41.8 (+3.7) |
| CE + CUDA Ahn et al. (2023) | 66.8 | 43.1 | - | 59.5 (+3.0) | 68.3 | 38.4 | 13.7 | 46.2 (+2.4) | 70.7 | 41.4 | 9.3 | 42.0 (+3.9) |
| CE + CMO + CUDA Ahn et al. (2023) | 65.7 | 43.0 | - | 58.7 (+2.2) | 67.6 | 39.2 | 17.6 | 47.0 (+3.2) | 71.1 | 43.4 | 11.7 | 43.6 (+5.5) |
| CE + DODA | 67.0 | 44.0 | - | 59.9 (+3.4) | 71.2 | 40.3 | 12.6 | 48.0 (+4.2) | 74.8 | 43.8 | 10.0 | 44.5 (+6.4) |
| CE + CMO + DODA | 67.1 | 42.5 | - | 59.5 (+3.0) | 70.4 | 41.2 | 19.7 | 49.3 (+5.5) | 73.2 | 44.4 | 12.4 | 44.9 (+6.8) |
| CE-DRW Cao et al. (2019) | 62.5 | 48.6 | - | 58.2 (+0.0) | 60.6 | 39.0 | 22.9 | 45.0 (+0.0) | 63.4 | 41.2 | 15.7 | 41.4 (+0.0) |
| CE-DRW + CUDA Ahn et al. (2023) | 64.2 | 56.2 | - | 61.7 (+3.5) | 63.8 | 48.0 | 37.0 | 52.5 (+7.5) | 63.5 | 48.9 | 25.3 | 46.9 (+5.5) |
| CE-DRW + DODA | 63.6 | 56.7 | - | 61.5 (+3.3) | 63.4 | 47.4 | 38.9 | 52.5 (+7.5) | 60.2 | 51.9 | 29.6 | 48.1 (+6.7) |
| LDAM-DRW Cao et al. (2019) | 62.7 | 46.1 | - | 57.5 (+0.0) | 63.0 | 41.2 | 25.1 | 47.2 (+0.0) | 62.8 | 42.6 | 21.1 | 43.2 (+0.0) |
| LDAM-DRW + CUDA Ahn et al. (2023) | 63.6 | 45.2 | - | 57.9 (+0.4) | 66.2 | 46.2 | 26.4 | 50.8 (+3.6) | 66.0 | 49.5 | 22.1 | 47.1 (+3.9) |
| LDAM-DRW + DODA | 63.3 | 45.8 | - | 57.9 (+0.4) | 64.7 | 46.3 | 27.5 | 50.5 (+3.3) | 65.4 | 50.8 | 25.5 | 48.3 (+5.1) |
| BS Ren et al. (2020) | 61.5 | 50.6 | - | 58.1 (+0.0) | 60.3 | 41.3 | 34.3 | 47.9 (+0.0) | 59.6 | 42.3 | 23.7 | 42.8 (+0.0) |
| BS + CUDA Ahn et al. (2023) | 64.2 | 55.5 | - | 61.5 (+3.4) | 63.6 | 48.4 | 37.3 | 52.7 (+4.8) | 62.5 | 49.1 | 29.4 | 47.9 (+5.1) |
| BS + DODA | 64.0 | 56.8 | - | 61.8 (+3.7) | 62.2 | 51.2 | 41.5 | 54.0 (+6.1) | 62.2 | 49.3 | 31.2 | 48.7 (+5.9) |
| RIDE (3 experts) Wang et al. (2021) | 66.4 | 49.4 | - | 61.1 (+0.0) | 65.7 | 47.7 | 31.8 | 52.2 (+0.0) | 65.7 | 48.6 | 25.0 | 47.5 (+0.0) |
| RIDE + CMO Park et al. (2022) | 65.3 | 48.5 | - | 60.1 (-1.0) | 67.8 | 47.0 | 33.4 | 53.1 (+0.9) | 67.9 | 51.2 | 27.6 | 50.0 (+2.5) |
| RIDE (3 experts) + CUDA Ahn et al. (2023) | 65.6 | 47.2 | - | 59.9 (-1.2) | 68.2 | 46.1 | 29.3 | 52.1 (-0.1) | 68.7 | 50.9 | 25.7 | 49.6 (+2.1) |
| RIDE (3 experts) + DODA | 65.7 | 49.9 | - | 60.8 (-0.3) | 67.0 | 46.5 | 33.6 | 52.6 (+0.4) | 68.4 | 51.1 | 27.8 | 50.2 (+2.7) |
| BCL Zhu et al. (2022) | 62.2 | 51.8 | - | 58.9 (+0.0) | 61.6 | 43.1 | 34.3 | 49.1 (+0.0) | 63.1 | 42.9 | 23.9 | 44.2 (+0.0) |
| BCL + CUDA Ahn et al. (2023) | 65.3 | 56.6 | - | 62.6 (+3.7) | 64.0 | 47.4 | 39.4 | 52.7 (+3.6) | 64.7 | 49.7 | 29.1 | 48.8 (+4.6) |
| BCL + DODA | 65.6 | 56.1 | - | 62.7 (+3.8) | 64.9 | 48.0 | 40.6 | 53.6 (+4.5) | 66.0 | 50.7 | 33.8 | 51.0 (+6.8) |

(2019), and iNaturalist 2018 Van Horn et al. (2018). CIFAR-100-LT and ImageNet-LT are long-tailed versions artificially truncated from the original balanced datasets, while iNaturalist 2018 is a real-world, naturally long-tailed dataset. CIFAR-100-LT has three imbalance ratio settings {10, 50, 100}, where the imbalance ratio is defined as $N_{max}/N_{min}$. For each dataset, we employ the officially provided version. The detailed descriptions for datasets are in **Appendix C**.

**Evaluation Metrics.** Model performance is mainly measured by the overall Top-1 accuracy (All). Following Ahn et al. (2023), we also statistically measure the accuracy on three disjoint subsets of the long-tailed datasets: head classes (Head), medium classes (Medium), and tail classes (Tail). Accuracy is reported as a percentage.

**Comparison Baselines.** We select a variety of long-tailed recognition methods as baselines, which are based on *different theoretical ideas*, including cross-entropy loss (CE) He et al. (2016), class re-balancing methods: CE-DRW Cao et al. (2019), LWS Kang et al. (2020), cRT Kang et al. (2020), LDAM-DRW Cao et al. (2019), Balanced Softmax (BS) Ren et al. (2020), information augmentation methods: CMO Park et al. (2022), CUDA Ahn et al. (2023), and module improvement methods: RIDE with three experts Wang et al. (2021), BCL Zhu et al. (2022). Thanks to the high flexibility of DODA, we integrate it with the aforementioned baseline to observe the gains DODA brings to existing methods. The detailed discussions and descriptions for baselines are in **Appendix B** and **Appendix C**.

**Implementation.** We use Pytorch Paszke et al. (2017) to implement all neural networks and train the model on 8 NVIDIA Tesla V100 GPUs. For CIFAR-100-LT dataset, we follow the general experimental setup from Cao et al. (2019) and utilize ResNet-32 He et al. (2016) as a backbone network. The networks are trained for 200 epochs by the SGD optimizer with an initial learning rate of $10^{-4}$, a momentum of 0.9, and a weight decay of $2 \times 10^{-4}$. We use a random strategy (up-weight and down-weight operations aren't active) for the first 50 epochs to avoid cold-boot issues and then switch to the proposed strategy for the remaining epochs. For ImageNet-LT and iNaturalist 2018 datasets, we use ResNet-50 as a backbone network. We train the network for 100 epochs using an initial learning rate of 0.1, and decay the learning rate at epochs 60 and 80 by 0.1. For all experiments, we set the hyperparameter values $p_{aug}$ as 0.5.

## 4.2 BENCHMARK RESULTS

**CIFAR-100-LT.** Table 1 displays the overall classification accuracies on CIFAR-100-LT dataset. It can be observed that DODA achieves comprehensive and stable improvements over the original long-tailed learning baselines: CE He et al. (2016), CE-DRW Cao et al. (2019), LDAM-DRW Cao et al. (2019), BS Ren et al. (2020), RIDE with three experts Wang et al. (2021), BCL Zhu et al.

Table 2: Accuracy (%) on ImageNet-LT and iNaturalist 2018 datasets wtih state-of-the-art methods.

| Method | ImageNet-LT | | | | iNaturalist 2018 | | | |
|---|---|---|---|---|---|---|---|---|
| | Head | Medium | Tail | All | Head | Medium | Tail | All |
| CE He et al. (2016) | 64.0 | 33.8 | 5.8 | 41.6 (+0.0) | 73.9 | 63.5 | 55.5 | 61.0 (+0.0) |
| CE + CUDA Ahn et al. (2023) | 67.1 | 47.1 | 13.4 | 47.2 (+5.6) | 74.6 | 65.0 | 57.2 | 62.5 (+1.5) |
| CE + DODA | **67.4** | 47.5 | 13.9 | 48.1 (+6.5) | **74.9** | 66.0 | 58.4 | 63.6 (+2.6) |
| CE-DRW Cao et al. (2019) | 61.7 | 47.3 | 28.8 | 50.1 (+0.0) | 68.2 | 67.3 | 66.4 | 67.0 (+0.0) |
| CE-DRW + CUDA Ahn et al. (2023) | 61.7 | 48.4 | 30.5 | 51.1 (+1.0) | 68.8 | 67.9 | 66.5 | 67.4 (+0.4) |
| CE-DRW + DODA | 62.4 | 48.5 | 31.3 | 52.2 (+2.1) | 69.0 | 68.2 | 67.8 | 68.2 (+1.2) |
| LWS Kang et al. (2020) | 57.1 | 45.2 | 29.3 | 47.7 (+0.0) | 65.0 | 66.3 | 65.5 | 65.9 (+0.0) |
| cRT Kang et al. (2020) | 58.8 | 44.0 | 26.1 | 47.3 (+0.0) | 69.0 | 66.0 | 63.2 | 65.2 (+0.0) |
| cRT + CUDA Ahn et al. (2023) | 62.3 | 47.2 | 28.1 | 50.2 (+2.9) | 68.2 | 67.9 | 66.4 | 67.3 (+2.1) |
| cRT + DODA | 62.8 | 47.7 | 28.9 | 51.3 (+3.6) | 69.2 | 68.3 | 67.6 | 68.5 (+3.3) |
| LDAM-DRW Cao et al. (2019) | 60.4 | 46.9 | 30.7 | 49.8 (+0.0) | - | - | - | 66.1 (+0.0) |
| LDAM-DRW + CUDA Ahn et al. (2023) | 63.2 | 48.2 | 31.2 | 51.5 (+1.7) | 68.0 | 67.5 | 66.8 | 67.3 (+1.2) |
| LDAM-DRW + DODA | 63.7 | 48.6 | 31.9 | 52.4 (+2.6) | 68.6 | 68.1 | 67.9 | 68.7 (+2.6) |
| BS Ren et al. (2020) | 60.9 | 48.8 | 32.1 | 51.0 (+0.0) | 65.7 | 67.4 | 67.5 | 67.3 (+0.0) |
| BS + CUDA Ahn et al. (2023) | 61.8 | 49.1 | 31.8 | 51.5 (+0.5) | 67.6 | 68.2 | 68.3 | 68.2 (+0.9) |
| BS + DODA | 61.9 | 49.5 | 32.4 | 52.0 (+1.0) | 68.1 | 68.9 | 69.5 | 69.4 (+2.1) |
| RIDE (3 experts) Wang et al. (2021) | 64.9 | 50.4 | 34.4 | 53.6 (+0.0) | 70.4 | 71.8 | 71.8 | 71.6 (+0.0) |
| RIDE + CMO Park et al. (2022) | 65.6 | 50.6 | 34.8 | 54.0 (+0.4) | 68.0 | 70.6 | 72.0 | 70.9 (-0.7) |
| RIDE (3 experts) + CUDA Ahn et al. (2023) | 66.0 | 51.7 | 34.7 | 54.7 (+1.1) | 70.6 | 72.6 | 72.7 | 72.4 (+1.4) |
| RIDE (3 experts) + DODA | 66.6 | 51.9 | 35.9 | 55.8 (+2.2) | 70.9 | 72.4 | **73.9** | **73.7 (+2.8)** |
| BCL Zhu et al. (2022) | 65.3 | 53.5 | 36.3 | 55.6 (+0.0) | 69.4 | 72.4 | 71.8 | 71.8 (+0.0) |
| BCL + CUDA Ahn et al. (2023) | 66.8 | 53.9 | 36.6 | 56.3 (+0.7) | 70.8 | 72.7 | 72.0 | 72.2 (+0.4) |
| BCL + DODA | 66.9 | **54.1** | **37.4** | **56.9 (+1.3)** | 71.2 | **73.2** | 73.4 | **73.7 (+1.9)** |

(2022). The outstanding performance on tail classes also demonstrates that DODA alleviates severe class sacrifice issues. Moreover, compared to existing class-independent long-tailed augmentation baselines: CMO Park et al. (2022) and CUDA Ahn et al. (2023), DODA achieves superior performance, especially on tail classes. Essentially, this improvement is due to the special 'care' given to the sacrificed classes in class-independent augmentation.

**ImageNet-LT and iNaturalist 2018.** We also compared DODA with state-of-the-art long-tailed recognition methods on large-scale datasets, and the experimental results are shown in Table 2. Applying DODA to the basic CE loss significantly improves the model performance and can be comparable to the performance of existing long-tailed learning methods. It is worth noting that ImageNet-LT and iNaturalist 2018 have higher imbalance ratios (i.e., 256 and 500) than CIFAR-100-LT (i.e., 10, 50, and 100), so the results also demonstrate that DODA can improve model performance in scenarios with varying degrees of imbalance.

## 4.3 FURTHER ANALYSIS

In this section, we conduct a detailed analysis of the mechanism of DODA and discuss the following four issues. All the analysis experiments are conducted on CIFAR-100-LT (IR=100). More empirical results are reported in **Appendix D**.

**Does DODA make fewer classes be 'sacrificed'?** In Section 2, we analyze and validate that class-independent DA is 'hypocritical', as it achieves the improvement of average performance by sacrificing certain classes (especially tail classes), which runs counter to the purpose of long-tailed learning. The goal of DODA is to pursue inter-class fairness while alleviating both inherent data-wise imbalance and extrinsic augmentation-wise imbalance. By allowing each class to choose appropriate DAs, DODA can effectively alleviate the problem of sacrificing 'weak' classes while achieving generalization performance. From the visualization of the accuracy of each class in Figure 4, it can be found that compared with CUDA,

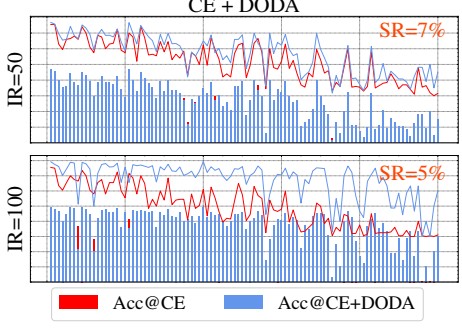

Figure 4: Visualization of the accuracy of each class between CE and CE with CUDA.

DODA reduces the sacrifice rate by 31% and 24%, respectively, indicating that DODA makes fewer classes be 'sacrificed'.

**Why DODA can alleviate the long-tailed problem?** Long-tailed learning aims to learn an accurate and robust deep model that can achieve generalized performance on long-tailed distribution tasks. DODA addresses the common problem of unfair DA in existing long-tailed methods from the perspective of DA. Our method is orthogonal to existing methods. By maintaining a preference list for each class, DODA provides each class with the right to choose DA fairly. As shown in Figure 5, we observe the trend of the selection hierarchies on two baselines (BS and CE). We randomly selected four classes (Index = {0, 10, 40, 80}) and 10 common DAs. It can be observed that, as the epoch increases, the preferred DA for specific classes gradually becomes apparent, indicating that each class can choose positive DAs to avoid being sacrificed, thereby alleviating the long-tailed problem.

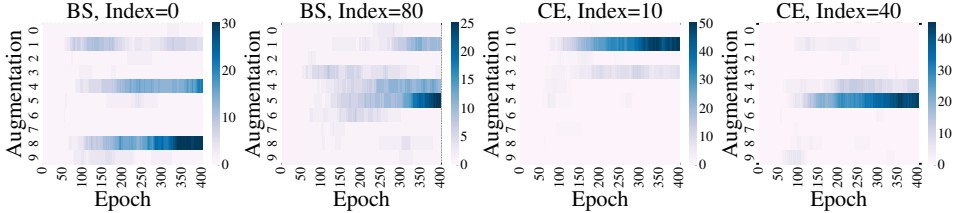

Figure 5: Trend of the selection hierarchies during training.

**What are the trends in DAs favored by classes?** As illustrated in Figure 6, we visualized the preferred augmentation methods for different classes, and it can be observed that certain augmentation methods are highly favored because some DAs tend to distort some decisive information in the data, while other DAs prefer to preserve label-related information.

**Would other augmentation methods be better?** We also compared DODA with existing DAs to demonstrate its superiority in long-tailed learning. Following the setting of Ahn et al. (2023), we select six augmentation methods, including AutoAugment (AA) Cubuk et al. (2019), Fast AutoAugment (FAA) Lim et al. (2019), DADA Li et al.

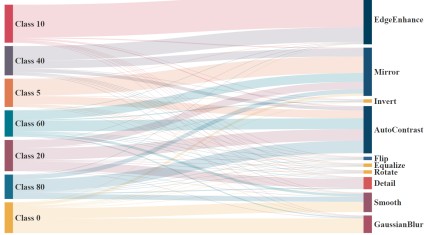

Figure 6: Class and 'its' preferred DA on BS and CIFAR-100-LT (IR=100).

(2020a), RandAugment (RA) Cubuk et al. (2020), and CUDA Ahn et al. (2023). Except for CUDA, other methods require additional computational resources to search for suitable DAs for datasets. Although CUDA implements class-wise strength adjustment, it still struggles to avoid class sacrifice issues. As shown in Figure 7, DODA outperforms other search-based and strength-based DAs, achieving fair augmentation while being computationally efficient.

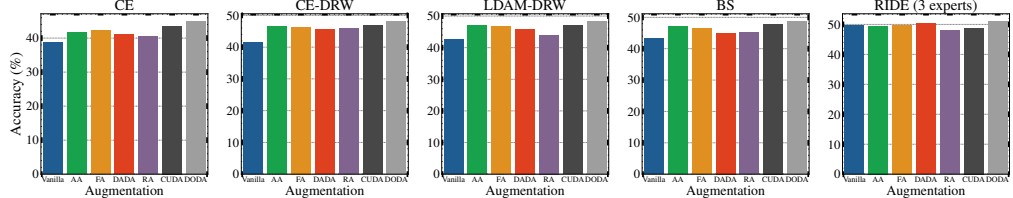

Figure 7: DA analysis on various algorithms, CE, CE-DRW, LDAM-DRW, BS, and RIDE.

## 5 CONCLUSION

In this study, we first theoretically analyzed the gains of traditional DAs in long-tailed learning and then proposed a 'stone' called Dynamic Optional Data Augmentation (DODA) to kill two 'birds': inherent data-wise imbalance and extrinsic augmentation-wise imbalance. DODA allows each class to choose appropriate DAs by maintaining a corresponding DA probability distribution for each class. Extensive experiments across long-tailed benchmarks verify the effectiveness of the DODA.

## 6 ACKNOWLEDGEMENT

This paper is partially supported by the National Natural Science Foundation of China (No.62072427, No.12227901), the Project of Stable Support for Youth Team in Basic Research Field, CAS (No.YSBR-005), Academic Leaders Cultivation Program, USTC.

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

# Appendix
# Kill Two Birds with One Stone:
# Rethinking Data Augmentation for Deep Long-tailed Learning

The content of the **Appendix** is summarized as follows:

1) in Sec. A, we state the proofs of Theorem 2 (Sec. 2.1), Theorem 3 (Sec. 2.2), and Theorem 4 (Sec. 3.2).

2) in Sec. B, we summarize existing long-tailed learning (LTL) and data augmentation (DA) methods and explicitly illustrate the novelty of DODA.

3) in Sec. C, we demonstrate the details of datasets and baselines we use in experiments of DODA.

4) in Sec. D, we illustrate more detailed empirical results and analyses of DODA.

## A    DETAILED PROOFS

### A.1    PROOF OF THEOREM 2

*Proof.* In Sec. 2.1, Theorem 1 states that when we approximate the ideal model $f^*$ by minimizing the training loss (i.e., 0 training error), the latter tends to zero, while the former is greater than zero due to the augmented samples deviate from the level set of $f^*$. Therefore, the trained model $f_\theta$ will be biased compared to the ideal model $f^*$.

$$\sum_{(x,y)\in\mathcal{D}} \mathbb{E}[||y - f^*(\mathcal{O}(x))||_2^2] > 0 \ \wedge \ \sum_{(x,y)\in\mathcal{D}} \mathbb{E}[||y - f_\theta(\mathcal{O}(x))||_2^2] = 0 \Longrightarrow \text{bias} \qquad (10)$$

From the perspective of the dataset, the reason for the bias is that the semantic information of some samples does not match their original labels after DA, which means that DA cannot guarantee that it is label-preserving. We reconsider this problem from the perspective of long-tailed learning. First of all, for the whole dataset $\mathcal{D}$, minimizing the training loss essentially means that the trained model $f_\theta$ should achieve 0 training error on each class, i.e.,

$$\sum_{(x,y)\in\mathcal{D}} \mathbb{E}[||y - f_\theta(\mathcal{O}(x))||_2^2] = 0 \Longrightarrow \sum_{(x,y)\in\mathcal{D}|y=c} \mathbb{E}[||y - f_\theta(\mathcal{O}(x))||_2^2] = 0 \quad \forall c \in C \qquad (11)$$

Similarly, for class $c$ and augmented samples $\{(\mathcal{O}(x), y)|y = c, (\mathcal{O}(x), y) \in \mathcal{D}\}$, when we achieve or approximately achieve the minimization of the training loss on class $c$, DA inevitably makes some samples of class $c$ mismatch with their original labels, i.e., the augmented samples deviate from the level set of the ideal model $f^*$. Therefore, when we use the augmented samples of class $c$ as inputs to $f^*$, $f^*$ cannot predict the labels completely correctly.

$$\sum_{(x,y)\in\mathcal{D}} \mathbb{E}[||y - f^*(\mathcal{O}(x))||_2^2] > 0 \Longrightarrow \sum_{(x,y)\in\mathcal{D}|y=c} \mathbb{E}[||y - f^*(\mathcal{O}(x))||_2^2] > 0 \quad \forall c \in C \qquad (12)$$

Although we achieve a seemingly optimal (ideal) training model $f_\theta$, its fitting process on class $c$ has actually deviated from the ideal optimization process. Therefore, the deviation between the trained model $f_\theta$ and the ideal model $f^*$ on class $c$ is inevitable, i.e., $f_\theta$ has class-wise bias.

$$\sum_{(x,y)\in\mathcal{D}|y=c} \mathbb{E}[||y - f^*(\mathcal{O}(x))||_2^2] > 0 \ \wedge \ \sum_{(x,y)\in\mathcal{D}|y=c} \mathbb{E}[||y - f_\theta(\mathcal{O}(x))||_2^2] = 0 \Longrightarrow \text{class-wise bias}$$

$$(13)$$

### A.2    PROOF OF THEOREM 3

*Proof.* According Definition 1, the distribution of each class in the training set can be approximate as a circle in a two-dimensional feature space, and the distribution center of class $c$ can be defined as $(\mathbb{X}_c, \mathbb{Y}_c)$ and the distribution radius is $\mathbb{R}_c$. So, the distribution span $\mathbb{S}_c$ can be expressed as follows:

$$\mathbb{S}_c \Rightarrow (\mathbb{X} - \mathbb{X}_c)^2 - (\mathbb{Y} - \mathbb{Y}_c)^2 = \mathbb{R}_c{}^2 \qquad (14)$$

For the data distribution $\mathbb{S}_{c_h}$ and $\mathbb{S}_{c_t}$ of head class $c_h$ and tail class $c_t$, the new distribution span after using uniform DA $\mathcal{O}(\cdot)$ can be defined as $\bar{\mathbb{S}}_{c_h} \Rightarrow (\mathbb{R}_{c_h} + \Delta_{c_h})^2$ and $\bar{\mathbb{S}}_{c_t} \Rightarrow (\mathbb{R}_{c_t} + \Delta_{c_t})^2$, and $\Delta_{c_h}$ and $\Delta_{c_t}$ represent the increase in distribution radius within each class after DA. For the same augmentation method, $\Delta_{c_h} = \Delta_{c_t}$.

Here, we define the original data distributions of head class $c_h$ and tail class $c_t$ as the base spaces $\mathbb{R}_{c_t}{}^2$ and $\mathbb{R}_{c_h}{}^2$, and define the expanded data distributions of head class $c_h$ and tail class $c_t$ as the marginal spaces $(\mathbb{R}_{c_t} + \Delta_{c_t})^2 - \mathbb{R}_{c_t}{}^2$ and $(\mathbb{R}_{c_h} + \Delta_{c_h})^2 - \mathbb{R}_{c_h}{}^2$

Then, the augmentation sensitivity of head class $c_h$ and tail class $c_t$ can be defined as $\psi_{c_h}$ and $\psi_{c_t}$.

$$\psi_{c_h} = \frac{(\mathbb{R}_{c_h} + \Delta_{c_h})^2 - \mathbb{R}_{c_h}{}^2}{\mathbb{R}_{c_h}{}^2} \tag{15}$$

$$\psi_{c_t} = \frac{(\mathbb{R}_{c_t} + \Delta_{c_t})^2 - \mathbb{R}_{c_t}{}^2}{\mathbb{R}_{c_t}{}^2} \tag{16}$$

Therefore, we can measure the augmentation sensitivity difference between head class $c_h$ and tail class $c_t$, i.e.,

$$
\begin{aligned}
\psi_{c_t} - \psi_{c_h} &= \frac{(\mathbb{R}_{c_t} + \Delta_{c_t})^2 - \mathbb{R}_{c_t}{}^2}{\mathbb{R}_{c_t}{}^2} - \frac{(\mathbb{R}_{c_h} + \Delta_{c_h})^2 - \mathbb{R}_{c_h}{}^2}{\mathbb{R}_{c_h}{}^2} \\
&= \frac{\mathbb{R}_{c_t}{}^2 + 2\mathbb{R}_{c_t}\Delta_{c_t} + \Delta_{c_t}{}^2 - \mathbb{R}_{c_t}{}^2}{\mathbb{R}_{c_t}{}^2} - \frac{\mathbb{R}_{c_h}{}^2 + 2\mathbb{R}_{c_h}\Delta_{c_h} + \Delta_{c_h}{}^2 - \mathbb{R}_{c_h}{}^2}{\mathbb{R}_{c_h}{}^2} \\
&= \frac{2\mathbb{R}_{c_t}\Delta_{c_t} + \Delta_{c_t}{}^2}{\mathbb{R}_{c_t}{}^2} - \frac{2\mathbb{R}_{c_h}\Delta_{c_h} + \Delta_{c_h}{}^2}{\mathbb{R}_{c_h}{}^2} \\
&= \frac{2\mathbb{R}_{c_t}\mathbb{R}_{c_h}{}^2\Delta_{c_t} + \mathbb{R}_{c_h}{}^2\Delta_{c_t}{}^2 - 2\mathbb{R}_{c_t}^2\mathbb{R}_{c_h}\Delta_{c_h} - \mathbb{R}_{c_t}{}^2\Delta_{c_h}{}^2}{\mathbb{R}_{c_t}{}^2\mathbb{R}_{c_h}{}^2} \\
&= \frac{2\mathbb{R}_{c_t}\mathbb{R}_{c_h}(\mathbb{R}_{c_h}\Delta_{c_t} - \mathbb{R}_{c_t}\Delta_{c_h})}{\mathbb{R}_{c_t}{}^2\mathbb{R}_{c_h}{}^2} + \frac{\mathbb{R}_{c_h}{}^2\Delta_{c_t}{}^2 - \mathbb{R}_{c_t}{}^2\Delta_{c_h}{}^2}{\mathbb{R}_{c_t}{}^2\mathbb{R}_{c_h}{}^2} \\
&= 2\Delta_{c_h} \cdot \frac{\mathbb{R}_{c_h} - \mathbb{R}_{c_t}}{\mathbb{R}_{c_h}\mathbb{R}_{c_t}} + \Delta_{c_h}{}^2 \cdot \frac{\mathbb{R}_{c_h}{}^2 - \mathbb{R}_{c_t}{}^2}{\mathbb{R}_{c_h}{}^2\mathbb{R}_{c_t}{}^2} \\
&> 0
\end{aligned} \tag{17}
$$

The above derivation indicating that tail classes are more sensitive to the marginal space.

For high-dimensional feature space,

*Proof.* Assuming that the dimension of high-dimensional features is $n$, the distribution center of class $c$ is defined as $(\mathbb{X}_c^1, \mathbb{X}_c^2, ..., \mathbb{X}_c^n)$ and the distribution radius is $\mathbb{R}_c$. The distribution span $\mathbb{S}_c$ can be expressed as

$$(\mathbb{X}^1 - \mathbb{X}_c^1)^2 - (\mathbb{X}^2 - \mathbb{X}_c^2)^2 - ... - (\mathbb{X}^n - \mathbb{X}_c^n)^2 = \mathbb{R}_c{}^2 \tag{18}$$

We assume that the data distribution of head class $c_h$ and tail class $c_t$ are $\mathbb{S}_{c_h}$ and $\mathbb{S}_{c_t}$ and the distribution span after DA $\bar{\mathbb{S}}_{c_h}$ and $\bar{\mathbb{S}}_{c_t}$. So the augmentation sensitivity $\psi$ of class $c$ can be expressed as follows:

$$\psi_c = \frac{\frac{\pi^{n/2}(\mathbb{R}_c + \Delta)}{\Gamma(1 + n/2)} - \frac{\pi^{n/2}\mathbb{R}_c}{\Gamma(1 + n/2)}}{\frac{\pi^{n/2}\mathbb{R}_c}{\Gamma(1 + n/2)}}, \tag{19}$$

and further deduce:

$$\psi_{c_t} - \psi_{c_h} = \frac{\pi^{n/2}}{\Gamma(1 + n/2)} \frac{(\mathbb{R}_{c_t}\mathbb{R}_{c_h} + \Delta\mathbb{R}_{c_h})^n - (\mathbb{R}_{c_t}\mathbb{R}_{c_h} + \Delta\mathbb{R}_{c_t})^n}{\mathbb{R}_{c_t}{}^n\mathbb{R}_{c_h}{}^n} > 0 \tag{20}$$

This indicates that this theoretical explanation is equally applicable to higher-dimensional spaces.

## A.3 PROOF OF THEOREM 4

*Proof.* We want to show that for a more dominant DA $\mathcal{O}^{k_1}$, the bias of $\mathcal{O}^{k_1}$ from $P$ is smaller than that of $\mathcal{O}^{k_2}$ from $P$, where $Q^{k_1}$ and $Q^{k_2}$ are the level-sets of the models $f_\theta^{k_1}$ and $f_\theta^{k_2}$ learned using DA $\mathcal{O}^{k_1}$ and DA $\mathcal{O}^{k_2}$, respectively, and $P$ is the level-set of the model $f_\theta$ trained on the original dataset.

Herr, we use Chebyshev's inequality to bound the probability that a random variable deviates from its expected value by a certain amount. Let $X$ be a random variable that represents the deviation of $f_\theta(x)$ from $y$ for a sample $(x, y)$ in the original dataset. Let $Y$ be a random variable that represents the deviation of $f_\theta^k(\mathcal{O}^k(x))$ from $y$ for a sample $(x, y)$ in the augmented dataset using DA $\mathcal{O}^k$. Then we have:

$$P(|X - E(X)| > t) \leq Var(X)/t^2 \tag{21}$$

$$P(|Y - E(Y)| > t) \leq Var(Y)/t^2 \tag{22}$$

The level-set bias $\delta(Q^k, P)$ can be defined as the degree of distributional deviation between the level-sets $Q^k$ and $P$. Intuitively, this can be measured by the difference between $E(Y)$ and $E(X)$, or the difference between $Var(Y)$ and $Var(X)$. We assume that $E(X) = 0$, since $f_\theta$ is trained to minimize the training loss on the original dataset. Then we have:

$$\delta(Q^k, P) = |E(Y)| + |Var(Y) - Var(X)| \tag{23}$$

Now, suppose that $\mathcal{O}^{k_1}$ dominates $\mathcal{O}^{k_2}$ on class $c$, i.e., $\nabla_{z_c^{k_1}}^{pos} > \nabla_{z_c^{k_2}}^{pos}$. This means that $f_\theta(\mathcal{O}^{k_1}(x))$ is more likely to be equal to $y$ than $f_\theta(\mathcal{O}^{k_2}(x))$ for samples $(x, y)$ in class $c$. Therefore, we have:

$$E(Y|y = c, \mathcal{O}^{k_1}) < E(Y|y = c, \mathcal{O}^{k_2}) \tag{24}$$

$$Var(Y|y = c, \mathcal{O}^{k_1}) < Var(Y|y = c, \mathcal{O}^{k_2}) \tag{25}$$

By taking the weighted average over all classes, we obtain,

$$E(Y|\mathcal{O}^{k_1}) < E(Y|\mathcal{O}^{k_2}) \tag{26}$$

$$Var(Y|\mathcal{O}^{k_1}) < Var(Y|\mathcal{O}^{k_2}) \tag{27}$$

Hence, we conclude that:

$$\delta(Q^{k_1}, P) < \delta(Q^{k_2}, P) \tag{28}$$

This completes the proof.

## B  RELATED WORK

### B.1  LONG-TAILED LEARNING (LTL)

Real-world training datasets typically exhibit a long-tailed class distribution, where a small fraction of classes have massive samples and the rest classes are associated with only a few samples. Unfortunately, the deep models trained by the common practice of empirical risk minimization cannot handle this distribution, resulting in a significant decrease in model performance Zhang et al. (2021b). Recently, missive novel longt-tailed learning methods have been proposed to learn a more generalized model from imbalanced training datasets, which can be divide into three main categories: class re-balancing Kang et al. (2020); Ren et al. (2020); Wang et al. (2020); Lin et al. (2017); Cui et al. (2019); Tan et al. (2020), module improvement Zhang et al. (2017b); Ouyang et al. (2016); Tang et al. (2020); Kang et al. (2020); Zhou et al. (2020); Zhang et al. (2022), and information augmentation Chu et al. (2020); Kim et al. (2020b); Hu et al. (2020); Zang et al. (2021); Park et al. (2022); Ahn et al. (2023).

Class re-balancing is the most typical strategy, which balances inter-class sample numbers or weights by re-sampling or cost-sensitive learning. On the one hand, traditional re-sampling methods, e.g., random over-sampling (ROS) and random under-sampling (RUS), achieve re-balancing by repeating the samples from tail classes and discarding the samples from head classes, but they tend to overfit

to tail classes when datasets are extremely unbalanced. To this end, recent studies propose class-balanced re-sampling strategies, e.g., bi-level class-balanced sampling Wang et al. (2020) and meta learning based sampling Ren et al. (2020). Besides from the perspective of classes, scheme-oriented sampling strategies try to re-balance classes by designing some specific learning schemes, such as quintuplet sampling Huang et al. (2016) and replay based sampling Kim et al. (2020a). On the other hand, some studies, called cost-sensitive learning, re-balance classes by adjusting the loss values of different classes. For example, CB Cui et al. (2019) proposed a effective number to approximate the expected sample number of each class, and Focal loss Lin et al. (2017) used the prediction probabilities to inversely re-weight classes.

In addition to class re-balancing, researchers also explored enhancing model performance by improving network modules. A intuitive method is decoupled training, which decouples the learning procedure into representation learning and classifier training. As a pioneering work, Decoupling Kang et al. (2020) proposed a two-stage training scheme and showed some refreshing observations. KCL Kang et al. (2021) and FRS Wang et al. (2023) believed that a balanced feature space is beneficial to LTL, so they designed contrastive learning based losses to learn a more class-balanced and class-discriminative feature space. Furthermore, as a classic theory, ensemble learning is also applied to LTL by designing and combining multiple expert networks. For instance, BBN Zhou et al. (2020) proposed to use two network branches to handle LTL. Following BBN, BAGS Li et al. (2020b) explored a multi-head scheme. Not restricted to a balanced test set, SADE Zhang et al. (2022) explored the multi-expert scheme to handle test distribution-agnostic LTL.

Although the overall performance is improved, these methods cannot essentially handle the issue of lacking information, particularly on tail classes due to limited data amount. Orthogonally, some information augmentation studies seek to introduce additional information into model training, such as FTL Yin et al. (2019) and M2m Kim et al. (2020b) transferred the knowledge from head classes to enhance model training on tail classes considering the inter-class knowledge imbalance. To solve information restrictions in essence, another line of research is to apply representation augmentation or data augmentation to LTL. For example, CMO Park et al. (2022) augmented diversified minority samples by leveraging the rich context of the majority classes as background images. Considering fairness, FSR Wang et al. (2023) and CUDA Ahn et al. (2023) advocate to find appropriate augmentation strength for each class. However, *although these methods enrich the overall information to a certain extent and improve model performance, they ignored the sacrifice of some classes behind this improvement*. **For this reason, we jointly pay attention to the inherent data-wise imbalance and extrinsic augmentation-wise imbalance, thereby minimizing the sacrifice.**

### B.2 DATA AUGMENTATION

DA has been applied in many fields because it can effectively alleviate overfitting and improve model generalization performance. DA is simple in design, and various DAs can be achieved through image manipulation, e.g., filp, crop, and rotate Robbins & Monro (1951). Recently, mixup based DA methods are proposed to improve model robustness by fusing two images and their labels Zhang et al. (2017a); Tokozume et al. (2018). Considering the diversity of DA, some studies try to combine them randomly or in order, such as AutoAugment Cubuk et al. (2019), Fast AutoAugment Lim et al. (2019), DADA Li et al. (2020a), and RandAugment Cubuk et al. (2020). In addition, researchers are improving DAs to make them suitable for LTL, however, they ignore that DA is class-independent, and thus may cause a mismatch between augmented data and actual labels Park et al. (2022); Wang et al. (2023); Ahn et al. (2023). Therefore, **it is necessary to design a class-dependent long-tailed DA to allow each class to choose an appropriate augmentation method.**

## C DATASET AND BASELINE DETAILS

### C.1 DATASET

**CIFAR-100-LT** Cao et al. (2019): is a long-tailed version of artificially truncated from the original balanced dataset CIFAR-100, which includes 100 different categories, 50,000 training images and 10,000 test images. The 100 categories in CIFAR-100 form 20 superclasses, each with 5 classes. CIFAR-100-LT has three imbalance ratio settings 10, 50, 100, where the imbalance ratio $\rho$ is defined as the ratio of the sample sizes of the most frequent and least frequent classes, i.e., $\rho = N_{max}/N_{min}$.

Table 3: Statistics of the long-tailed datasets.

| Dataset | # of Classes | # of Training set | # of Test set | Imbalance ratio |
|---|---|---|---|---|
| CIFAR-100-LT | 100 | 50,000 | 10,000 | {10, 50, 100} |
| ImageNet-LT | 1,000 | 115,846 | 50,000 | 256 |
| iNaturalist 2018 | 8,142 | 437,513 | 24,426 | 500 |

**ImageNet-LT** Liu et al. (2019): is a long-tailed version of artificially truncated from the original balanced dataset ImageNet, which includes 1,000 different categories, 115,846 training images and 50,000 test images. The most frequent or least frequent class has 1,280 or 5 images, so the imbalance ratio $\rho = 256$.

**iNaturalist 2018** Van Horn et al. (2018): is a real-world, naturally long-tailed dataset, which includes 8,142 different categories, 437,513 training images and 24,426 test images. Each image has one ground truth label. The iNat dataset is highly imbalanced with dramatically different number of images per category and the imbalance ratio $\rho$ is 500.

## C.2 AUGMENTATION

we incorporated ten commonly used DA methods in our experiments, and descriptions are shown in Table 4. The specific code implementation can be found in '/aug/doda.py'.

Table 4: Description of DAs utilized in DODA.

| DA | Parameter | Description |
|---|---|---|
| Flip | 0/1 | Flip top and bottom |
| Mirror | 0/1 | Flip left and right |
| EdgeEnhance | 0/1 | Increasing the contrast of the pixels around the targeted edges |
| Detail | 0/1 | Utilize convolutional kernel [[0,-1, 0], [-1, 10,-1], [0,-1, 0]] |
| Smooth | 0/1 | Utilize convolutional kernel [[1, 1, 1], [1, 5, 1], [1, 1, 1]] |
| AutoContrast | 0/1 | Remove a specific percent of the lightest and darkest pixels |
| Equalize | 0/1 | Apply non-linear mapping to make uniform distribution |
| Invert | 0/1 | Negate the image |
| GaussianBlur | [0, 2] | Blurring an image using Gaussian function |
| Rotate | [0, 30] | Rotate the image |

## C.3 BASELINES

To ensure a fair comparison, we select a large number of long-tailed learning methods as baselines in our experiments, and integrate DODA with these baselines to evaluate the effectiveness and flexibility of DODA. In addition, we also select the state-of-the-art data augmentation methods as comparison baselines to prove the superiority of DODA in long-tailde learning.

**Long-tailed methods:**

- CE He et al. (2016): is a cross-entropy loss based model, which is one of the most classic methods in the field of deep long-tailed learning.
- CE-DRW Cao et al. (2019): is a two-stage fine-tuning strategy based on cross-entropy loss.
- LWS Kang et al. (2020): is a two-stage training strategy, which keeps both the representations and classifier weights fixed and only learn the scaling factors.

- cRT Kang et al. (2020): is a two-stage training strategy, which keeps the representations fixed and randomly re-initialize and optimize the classifier weights using class-balanced sampling.

- LDAM-DRW Cao et al. (2019): extends the existing soft margin loss by enforcing class-dependent margins based on label frequencies and further introduces a deferred re-balancing optimization schedule.

- BS Ren et al. (2020): proposes to use the label frequencies to adjust mode predictions during training, so that the bias of class imbalance can be alleviated by the prior knowledge.

- RIDE (3 experts) Wang et al. (2021): introduces a knowledge distillation multi-expert framework to reduce the parameters by learning a student network with fewer experts.

- BCL Zhu et al. (2022): proposes a balanced contrastive learning loss and learns stronger feature representations through a dual-branch framework.

- CMO Park et al. (2022): focuses on utilizing the rich context of majority samples to improve the diversity of minority samples and mixes minority and majority images by using CutMix to enhance balancing and robustness simultaneously.

- CUDA Ahn et al. (2023): is a simple and efficient curriculum, which is designed to find the appropriate per-class strength of data augmentation.

**Data augmentation methods:**

- AutoAugment Cubuk et al. (2019): describes a simple procedure to automatically search for improved data augmentation policies by designing a search space where a policy consists of many sub-policies, one of which is randomly chosen for each image in each mini-batch.

- Fast AutoAugment Lim et al. (2019): finds effective augmentation policies via a more efficient search strategy based on density matching.

- DADA Li et al. (2020a): relaxes the discrete DA policy selection to a differentiable optimization problem via Gumbel-Softmax and introduces an unbiased gradient estimator to learn an efficient and accurate DA policy.

- RandAugment Cubuk et al. (2020): proposes a simplified search space that vastly reduces the computational expense of automated augmentation, and permits the removal of a separate proxy task.

## D  MORE EMPIRICAL RESULTS

### D.1  PARAMETER SENSITIVITY ANALYSIS OF $p_{aug}$

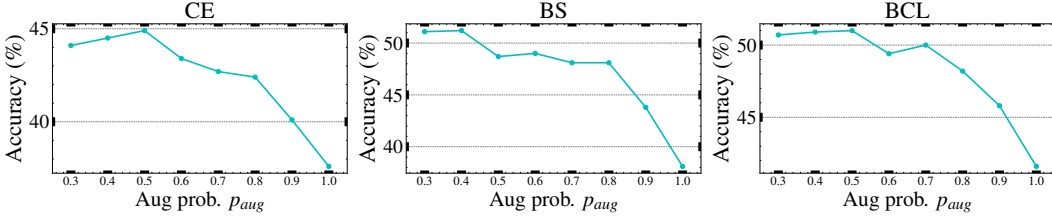

Figure 8: Parameter sensitivity analysis of augmentation probability $p_{aug}$ on CE, BS, and BCL.

To preserve the knowledge of the original dataset, we define the augmentation probability $p_{aug}$. For further analyze the impact of $p_{aug}$, we conduct a sensitivity analysis of hyperparameter $p_{aug}$. As shown in Figure 8, we test 8 different hyperparameter settings on three baselines, and the experimental results showed that a too small augmentation probability cannot sufficiently improve the model's generalization, while a too large augmentation probability cannot retain the knowledge in the original dataset, resulting in a decrease in model performance.

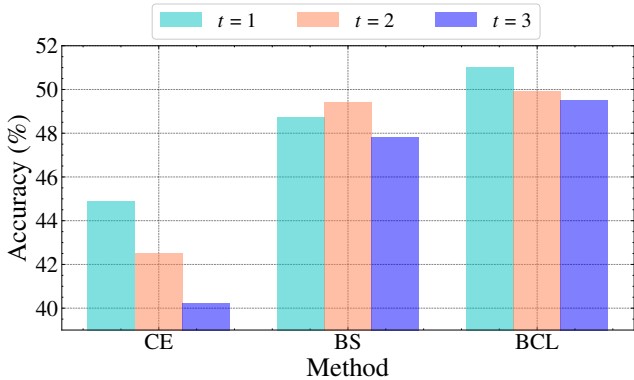

Figure 9: Parameter sensitivity analysis of number of DAs be selected $t$.

## D.2 Parameter Sensitivity Analysis of $t$

In previous analyses, we select the optimal DA for each class. However, we find that in some baselines, multiple DAs can be beneficial. Therefore, we further conduct a parameter sensitivity analysis on the number of DAs be selected $t$. As shown in Figure 9, we test three hyperparameter settings ($t = 1, 2, 3$) on three baselines. It can be observed that on CE and BCL, the model tends to select the optimal DA, while on BS, it tends to select both the optimal and suboptimal DAs. This phenomenon is consistent with the trend of selection hierarchies during training mentioned in the main text.

## D.3 Parameter Sensitivity Analysis of Num. of DAs

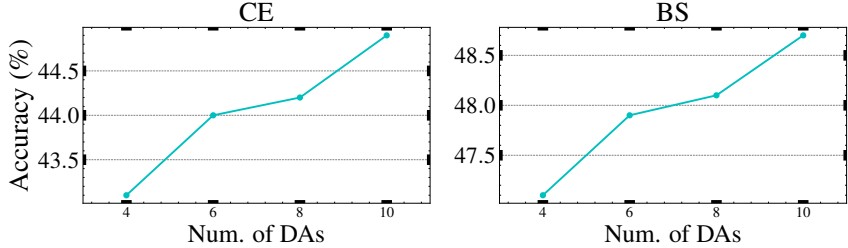

Figure 10: Impact of different numbers of DAs CIFAR-100-LT (IR=100).

As shown in Figure 10, we gradually reduced the number of DAs based on the degree of preference. The results indicate that reducing the number of augmentations leads to a loss of diversity. However, when the 'neglected' augmentations are removed, the model performance does not significantly degrade.

## D.4 Network Architecture Analysis

As shown in Figure 11, following Ahn et al. (2023), we also utilize ResNet-10 Liu et al. (2019) and ResNeXt-50 Xie et al. (2017) as our backbone network on ImageNet-LT. We conduct comparative experiments on three baselines (e.g., CE, BS, and BCL), and the results show that no matter what kind of backbone is used, DODA can always bring stable improvement to long-tailed learning algorithms.

## D.5 Training Time Analysis

In DODA's augmentation pipeline, we require additional computations to update and maintain the augmentation preference list for each class. Therefore, compared to the original baselines, using

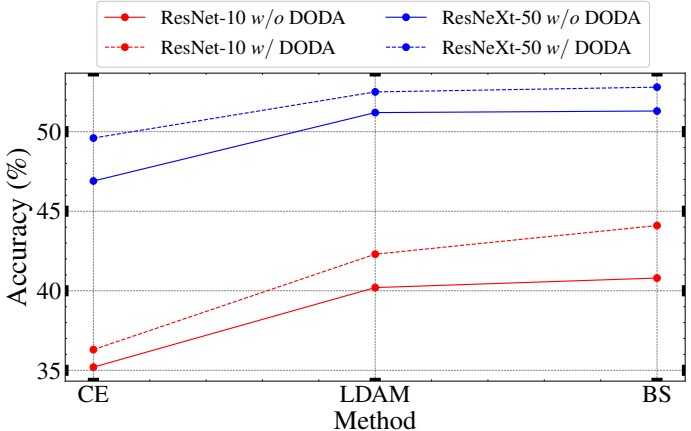

Figure 11: Network architecture analysis.

DODA incurs additional training time. As shown in Table 5, using DODA inevitably brings varying degrees of computational cost, but these costs are acceptable. For example, BS *w/o* DODA achieves better model performance and avoids serious sacrifices with only $\times$ 0.09 additional cost.

Table 5: Training time (min.) analysis on various algorithms.

| Method | CE | BS | BCL |
|---|---|---|---|
| *w/o* DODA | 60 | 68 | 94 |
| *w/* DODA | 68 ($\times$ 1.13) | 74 ($\times$ 1.09) | 102 ($\times$ 1.09) |

### D.6 MORE ANALYSIS ON SACRIFICE RATES

**SR on Different Long-tailed Baselines:** We provided the sacrifice rates of different data augmentations on CE in Figure 2, indicating that DA can lead to sacrifice problems for the original baseline. Similarly, long-tailed learning baselines also face this issue. Based on your comments, we have conducted further experiments on cRT and CIFAR-100-LT dataset (IR = 100). The results in Table 6 show that CUDA improved accuracy while sacrificing performance for certain classes, while DODA mitigated this sacrifice issue by preserving performance across classes.

Table 6: Accuracy (%) on CIFAR-100-LT dataset (IR = 100) wtih cRT. SR (%) indicates the sacrifice rate.

| Method | Head | Medium | Tail | All | SR |
|---|---|---|---|---|---|
| CE | 65.6 | 36.2 | 8.2 | 38.1 | - |
| CE + CUDA | 70.7 | 41.4 | 9.3 | 42.0 | 29 |
| CE + DODA | 74.8 | 43.8 | 10.0 | 44.5 | 5 |
| cRT Kang et al. (2020) | 64.4 | 49.1 | 25.8 | 47.5 | - |
| cRT + CUDA | 63.2 | 50.9 | 26.6 | 47.9 | 22 |
| cRT + DODA | 64.4 | 51.2 | 27.5 | 48.7 | 6 |

In general, just like focusing on tail classes when improving the average accuracy, when applying DAs in long-tailed learning, focusing on vulnerable classes that are easy to be sacrificed is also in line with the purpose of long-tailed learning.

**SR on Different DA Baselines:** We also tested different class-independent techniques (e.g., AutoAugment, CutOut) to demonstrate the superiority of our method. The specific experimental results are shown in Table 8.

AutoAugment improves the average accuracy on cRT and takes effect on each shot. However, we further analyze the sacrifice problem caused by DAs, and we find that despite achieving good performance, AutoAugment still cannot avoid the sacrifice problem, which means,

- The performance improvement of AutoAugment is hypocritical, for example, in the tail classes, the model achieves performance gains on some classes, while performing badly on others (i.e., pleasing the 'strong' and bullying the 'weak'). This sacrifice goes against the purpose of long-tailed learning despite the average performance improvement of the model on the tail classes.

- Both class-independent techniques lead to the sacrifice problem of sacrifice. From the sacrifice rate of different shots, it can be found that compared with the head classes, more classes in the tail classes are sacrificed, indicating that the tail classes are more likely to be regarded as the bullied 'weak' mentioned above.

Table 7: Accuracy (%) on CIFAR-100-LT dataset (IR = 100) wtih cRT. (·) indicates the sacrifice rate of different shots.

| Method | Head | Medium | Tail | All |
|---|---|---|---|---|
| cRT Kang et al. (2020) | 64.4(-) | 49.1(-) | 25.8(-) | 47.5(-) |
| cRT + AutoAugment | 64.8(5) | 49.9(6) | 25.9(13) | 47.9(24) |
| cRT + CutOut | 61.3(12) | 44.5(15) | 21.7(23) | 43.6(50) |
| cRT + DODA | 64.4(2) | 51.2(1) | 27.5(3) | 48.7(6) |

**SR on Different Epochs:** Here, we analyzed the changes in the sacrifice rate. The results shown in Table 2 show that the sacrifice problem caused by previous DAs cannot be eliminated during training, while DODA significantly improves this.

Table 8: Sacrifice rate (%) on Various Epochs.

| Epoch | 100 | 200 | 400 |
|---|---|---|---|
| CE + RandAugment | 328 | 315 | 309 |
| CE + DODA | 73 | 59 | 55 |

## D.7 More Comparisons with Modified Two-stage Model

Here, we compare DODA with CC-SAM Zhou et al. (2023), which is a two-stage model improvement method that trains the model in a decoupled manner and introduces class-conditional sharpness-aware minimization in the first stage. We have improved the existing open-source implementation and incorporated DODA's augmentation strategy. The quantitative experimental results are shown in Table 9.

Table 9: Accuracy (%) on CIFAR-100-LT dataset (Imbalance ratio = 100) wtih CC-SAM. SR (%) indicates the sacrifice rate.

| Method | Head | Medium | Tail | All | SR |
|---|---|---|---|---|---|
| CC-SAM Zhou et al. (2023) | 67.6 | 51.2 | 30.5 | 50.7 (+ 0.0) | - |
| CC-SAM + CUDA Ahn et al. (2023) | 67.5 | 52.0 | 30.7 | 51.0 (+ 0.3) | 31 |
| CC-SAM + DODA | 68.4 | 53.7 | 33.6 | 52.8 (+ 3.1) | 6 |

### D.8 MORE COMPARISONS WITH AUTO DA IN OTHER FIELDS

In this section, we compare DODA with Auto DA selection algorithms in other fields. Here we choose the most advanced method Zaiem et al. (2022) in the speech field as a comparison. However, directly applying the complete method from it in long-tailed learning does not lead to fair comparisons. So we partially implemented the augmentation strategies proposed in Zaiem et al. (2022). Firstly, since Zaiem et al. (2022) relies on a carefully designed pretext task, we replaced it with contrastive learning using cropping and augmentation, where pretext labels for each augmented view of a sample corresponding to the ID of the sample it originated from. Then, we replaced the downstream task related to speech with a long-tailed classification task. The experimental results of this modified implementation are shown in Table 10.

It can be observed that using the automatic augmentation strategy from Zaiem et al. (2022) results in limited performance improvement, while our method outperforms it significantly. The reasons for this are as follows: (1) Zaiem et al. (2022) relies on a carefully designed pretext task, so the improvement it brings may come from diversified data augmentation. (2) Zaiem et al. (2022) lacks the necessary focus on the tail classes, while our method pays more attention to inter-class fairness, resulting in better performance.

Table 10: Accuracy (%) on CIFAR-100-LT dataset (Imbalance ratio = 100) wtih Zaiem et al. (2022).

| Method | Head | Medium | Tail | All |
|---|---|---|---|---|
| CE | 65.6 | 36.2 | 8.2 | 38.1 (+ 0.0) |
| CE + Zaiem et al. (2022) | 68.9 | 38.7 | 8.4 | 40.2 (+ 2.1) |
| CE + DODA | 74.8 | 43.8 | 10.0 | 44.5 (+ 6.4) |

### D.9 EXPLORATION OF COMBINATIONS WITH SOTA LONG-TAILED DA

From the macro perspective of long-tailed learning, both DODA and CUDA belong to dynamic DA. However, at the methodology level, the two are different. A simple comparison is as Table 11:

Table 11: Comparison at the methodology level.

| Method | Adaptive Strength | Adaptive Function | Inter-class Fairness | Cold-boot Issues |
|---|---|---|---|---|
| CUDA | ✓ | | | |
| DODA | | ✓ | ✓ | ✓ |

It can be seen that to ensure fairness between classes while improving accuracy, we have made some methodology-level improvements. More interestingly, we find that CUDA and DODA are orthogonal, and we can find the optimal DA function and strength at the same time. The exploratory results are as follows:

Table 12: Exploratory results (%) on CIFAR-100-LT dataset (IR = 100).

| Method | Head | Medium | Tail | All |
|---|---|---|---|---|
| CE + DODA | 74.8 | 43.8 | 10.0 | 44.5 |
| CE + DODA + CUDA | 74.7 | 44.1 | 10.2 | 44.6 |

Although the performance gain is limited, continuing to explore this compositionality is beneficial for long-tailed learning.

## D.10 MORE TRENDS OF THE SELECTION HIERARCHIES ON DIFFERENT INDEXES

Figure 12: Trends of the selection hierarchies on different indexs.

