# OpenReview forum: "Kill Two Birds with One Stone: Rethinking Data Augmentation for Deep Long-tailed Learning"
_ICLR.cc/2024/Conference — ICLR 2024 poster_

### Official Review · Reviewer_rn58 · 2023-10-15

**Soundness:** 4 excellent
**Presentation:** 3 good
**Contribution:** 3 good
**Rating:** 8
**Confidence:** 4

**Summary:**

This study focus on data augmentation for long-tailed learning. First, the authors theoretically analyzes that the traditional data augmentation techniques cause  the long-tailed distribution to be imbalanced again. Specifically, the traditional data augmentation techniques cause inherent data-wise imbalance and extrinsic augmentation-wise imbalance, which are called as two birds in this paper. To kill the two birds, the authors propose an data augmentation technique termed DODA which make each class to choose appropriate augmentation methods. Experimental results on 4 benchmark datasets verify effectiveness of the DODA.

**Strengths:**

Code is submitted.

Many studies are reviewed in the Appendix.

The paper is easy to read.

Theoretical ground for the proposed algorithm.

Extensive experiments are conducted with 4 benchmark datasets.

Detailed analyses for the superiority of the proposed algorithm are provided.

The proposed algorithm can be orthogonally used with other long tailed learning algorithm such as LDAM and RIDE.

**Weaknesses:**

I cant find weakness of this paper.

**Questions:**

I have no question.

---

> ### Author Response · Authors · 2023-11-19
> **Reply to Reviewer rn58**
>
> Thanks so much for your time and positive feedback! We are encouraged and will continue to polish our work.

---

> ### Public Comment · ~Claude_Ross1 · 2024-07-31
> **Thank you**
>
> Thank you for your thorough review and positive feedback. Your insights on the strengths and the soundness of the theoretical and experimental work are valuable https://geometry-dashonline.com. Glad you found no weaknesses!

---

### Official Review · Reviewer_ZxxQ · 2023-10-31

**Soundness:** 3 good
**Presentation:** 3 good
**Contribution:** 3 good
**Rating:** 6
**Confidence:** 3

**Summary:**

Data augmentation (DA) is one of the strategies employed to address the dataset imbalance issue in long-tailed classification. This paper proposes the DODA algorithm based on the premise that in situations handling long-tailed datasets, the effectiveness of DA may vary for each class. In a nutshell, the proposed algorithm includes setting up a set of DAs and adjusting the preferences for each augmentation individually for each class during the training process. The paper offers experimental findings for CIFAR-LT, ImageNet-LT, and iNaturalist-2018.

**Strengths:**

1. The provided problem regarding data augmentation in long-tailed classification has a well-established and clear motivation.
2. Comprehensive experimental outcomes are presented, encompassing a range of existing training approaches and data augmentation techniques for addressing long-tailed classification.

**Weaknesses:**

1. The straightforward yet most important baseline is missing. The central idea presented in this paper is that there are both positive and negative DA effects specific to each class when working with long-tailed datasets, and it is beneficial to adjust them during the training process through the proposed DODA algorithm. Consequently, keeping the per-class weight distribution $\mathfrak{Q}_c$ fixed as a uniform distribution in the proposed DODA algorithm would serve as an essential baseline, which highlights the efficacy of the proposed adjusting approach. One can interpret the performance improvement described in the paper as simply due to the incorporation of extra data augmentation.

2. While the main tables present average values, the standard deviations are missing; it would be better to provide complete results, including standard deviations, in the appendices.

**Questions:**

1. Could you provide some valuable insights regarding the decline in performance that we see in the RIDE column of Table 1?
2. UniformAugment (LingChen et al., 2020) is a suitable baseline for comparison because of its straightforward application of a set of augmentations at random. It would be nice to see the UniformAugment baseline results.
3. It seems that the proposed methodology could be effective even in scenarios handling balanced datasets. Have you experimentally verified this by any chance?
4. ’its’ in the caption of Figure 6; quotation mark issues in LaTeX :)

---
LingChen et al., 2020, UniformAugment: A Search-free Probabilistic Data Augmentation Approach.

---

> ### Author Response · Authors · 2023-11-19
> **Reply to Reviewer ZxxQ (I)**
>
> We appreciate your comments. To address your concerns, below we conduct further comparative experiments and provide point-to-point responses. We will carefully revise our paper by taking into account all your suggestions. Looking forward to discussing more with you.
>
> > **Comment 1: Further Comparisons with UniformAugment. -** "UniformAugment (LingChen et al., 2020) is a suitable baseline for comparison because of its straightforward application of a set of augmentations at random. It would be nice to see the UniformAugment baseline results."
>
> Thank you for your insightful comments. Based on your suggestions, we have conducted comparative experiments on UniformAugment and DODA on CIFAR-100-LT (IR = 50 and 100). The experimental results shown in the table below indicate that under imbalanced distribution, DODA achieves better performance than UniformAugment. This improvement is comprehensive, especially in the tail classes, indicating that allowing each class to choose the augmentation method that is beneficial to itself can avoid significant sacrifices. This is consistent with what you mentioned.
>
> **Table: Accuracy (%) on CIFAR-100-LT dataset (IR = 50 and 100) wtih UniformAugment.**
> |Method|Head|Medium|Tail|All (50)|Head|Medium|Tail|All (100)|
> |:-|:-:|:-:|:-:|:-:|:-:|:-:|:-:|:-:|
> |CE + UniformAugment|68.5|34.8|10.4|**44.2**|71.5|38.5|5.8 |**40.2**|
> |CE + DODA          |71.2|40.3|12.6|**48.0**|74.8|43.8|10.0|**44.5**|
> |BS + UniformAugment|62.7|46.7|38.0|**51.5**|62.8|49.1|26.7|**47.2**|
> |BS + DODA          |62.2|51.2|41.5|**54.0**|63.1|49.3|31.2|**48.7**|
>
> > **Comment 2: Results with Standard Deviations. -** "While the main tables present average values, the standard deviations are missing; it would be better to provide complete results, including standard deviations, in the appendices."
>
> Thank you for your comments. Here, we simply provide a demonstrative result. The table shows partial results of DODA on CIFAR-100-LT, including standard deviations. Furthermore, based on your suggestions, we will supplement other results in the appendix to ensure the comprehensiveness of our work.
>
> **Table: Accuracy (%) on CIFAR-100-LT dataset.**
> |Method|IR = 10|IR = 50|IR = 100|
> |:-|:-:|:-:|:-:|
> |CE  + DODA|59.9 $\pm$ 1.2|48.0 $\pm$ 0.8|44.5 $\pm$ 0.5|
> |BS  + DODA|61.8 $\pm$ 1.0|54.0 $\pm$ 0.9|48.7 $\pm$ 0.7|
> |BCL + DODA|62.7 $\pm$ 1.0|53.6 $\pm$ 0.4|51.0 $\pm$ 0.6|

---

> ### Author Response · Authors · 2023-11-19
> **Reply to Reviewer ZxxQ (II)**
>
> > **Comment 3: Explanation of the performance of RIDE. -** "Could you provide some valuable insights regarding the decline in performance that we see in the RIDE column of Table 1?"
>
> Thank you for your comments. As you mentioned, we find that the performance of DODA in balanced settings or settings close to balanced (IR = 10) is not as stable as in imbalanced settings, such as fluctuating performance on RIDE. We consider that its multi-expert setting may lead to inconsistencies in the tendency when choosing augmentations, i.e., conflicting tendencies. Nevertheless, by observing the experimental results, we find that DODA still shows superior performance in the medium classes.
>
> > **Comment 4: More Experiments on balanced datasets. -** "It seems that the proposed methodology could be effective even in scenarios handling balanced datasets. Have you experimentally verified this by any chance?"
>
> Thank you for your comments. We have conducted exploratory experiments on balanced data distribution, e.g., original CIFAR-100, but it was not mentioned in our paper because our current focus is on the long-tailed setting. We strongly agree with your point of view, and the experimental results also confirm this. From the results in the table below, we can see that DODA still shows some effectiveness under balanced settings, although it is not as good as some methods for balanced distribution. Considering that DODA is aimed at the sacrifice problem under imbalanced distribution, we think this performance is acceptable.
>
> **Table: Accuracy (%) on CIFAR-100 dataset.**
> |Method|AutoAugment|UniformAugment|CUDA|DODA|
> |:-:|:-:|:-:|:-:|:-:|
> |Acc.|71.6|69.2|69.8|69.6|
>
> The original baseline uses Wide-ResNet as the backbone. To maintain fairness, we replace it with a standard ResNet-32.
>
> > **Comment 5: Typos.**
>
> Thanks. We sincerely appreciate and promise to revise thoroughly.

---

> > ### Comment · Reviewer_ZxxQ · 2023-11-19
> > **Response to authors**
> >
> > I appreciate the authors' dedication to conducting additional experiments with UniformAugment. My follow-up question is whether UniformAugment employed the augmentations specified in Table 4 of DODA (or a set of augmentations listed in LingChen et al. (2020)).

---

> > > ### Author Response · Authors · 2023-11-20
> > > **Further Response to Reviewer ZxxQ**
> > >
> > > Thank you very much for your timely response. Following your suggestions, to achieve a fair comparison, we used the same augmentation list as DODA in the implementation of UniformAugment. In this way, we can more intuitively understand our method.

---

> > > > ### Comment · Reviewer_ZxxQ · 2023-11-20
> > > >
> > > > Sounds great! I believe the additional baseline of keeping a uniform per-class weight distribution in the proposed DODA algorithm effectively highlights the efficacy of the suggested DODA approach. Given that the authors have addressed my primary concern regarding this baseline, I am increasing my score to 6 accordingly.

---

> > > > > ### Author Response · Authors · 2023-11-20
> > > > > **Thanks again!**
> > > > >
> > > > > Thanks again for your valuable time and insightful follow-up comments!

---

> ### Comment · Reviewer_ZxxQ · 2023-11-22
>
> I have an additional comment regarding Section 4.3. The paper said,
> > From the visualizations of the accuracy of each class in Figure 4, it can be found that compared with CUDA, DOA reduces the sacrifice rate by 31% and 24%, respectively, indicating that DODA makes fewer classes be `sacrificed.'
>
> It seems that the authors' obtained `31%` and `24%` by computing `31% = 38% (of CE + CUDA in Figure 2) - 7% (of CE + DODA in Figure 4)` and `24% = 29% (of CE + CUDA in Figure 2) - 5% (of CE + DODA in Figure 4)`, as they replied to Reviewer PqHc. However, when expressing the arithmetic difference between two percentages, `%p` should be used instead of `%`. Please make this correction; `31%` to `31%p` and `24%` to `24%p`.

---

> > ### Author Response · Authors · 2023-11-23
> >
> > Thanks again for your time and thoughtful suggestions, we have corrected this error :)

---

### Official Review · Reviewer_rRZN · 2023-11-01

**Soundness:** 2 fair
**Presentation:** 3 good
**Contribution:** 2 fair
**Rating:** 5
**Confidence:** 5

**Summary:**

This paper addresses the problem of imbalance in existing data augmentation techniques in long tailed learning. This paper proposes to improve this by introducing a dynamic augmentation strategy - DODA (Dynamic Optional Data Augmentation) to reinforce well-performing augmentation strategies and punish the inefficient ones during training. The authors claim that this method kills two ‘birds’ - inherent data imbalance and extrinsic augmentation wise imbalance with one ‘stone’.

**Strengths:**

The work addresses the problem of long tailed learning from a data augmentation perspective and shows potential for huge gains by simply using augmentations efficiently without much computational overhead.

The paper has a simple writing style and is easy to read and follow.

**Weaknesses:**

**Proposed Method**

a) The proposed method for DODA is exclusively used where augmentations  are solely for one single class and do not encourage inter class interaction. Augmentation methods such as mixup and its improved version Remix (Chou et al. 2020) have proven to be efficient in data-imbalance settings. However, comparison of the proposed method with Remix + LTL methods is lacking.
The algorithm optimizes the augmentation strategy on 10 predefined augmentation functions with fixed strengths. The authors should discuss the optimality of the selection of these functions and provide ablation studies for the same.

b) The algorithm optimizes the augmentation strategy on 10 predefined augmentation functions with fixed strengths. The authors should discuss the optimality of the selection of these functions and provide ablation studies for the same.

**Missing Baselines **

The effectiveness of DODA on state-of-the-art long-tailed methods PaCo [R1], and NCL[R2] is not demonstrated in the paper. It would be great if the results are provided in the rebuttal.

[R1] Cui, Jiequan et. al. “Parametric Contrastive Learning”, Proceedings of the IEEE/CVF international conference on computer vision, 2021.

[R2] Li, Jun et. al. “Nested Collaborative Learning for Long-Tailed Visual Recognition”. Proceedings of the IEEE/CVF international conference on computer vision, 2022.

**Questions:**

For contrastive methods such as BCL, how is DODA being used to generate multiple views of the same image for different classes?

---

> ### Author Response · Authors · 2023-11-19
> **Reply to Reviewer rRZN (I)**
>
> We appreciate your comments. To address your concerns, below we conduct further comparative experiments and provide point-to-point responses. We will carefully revise our paper by taking into account all your suggestions. Looking forward to discussing more with you.
>
> > **Comment 1: Further Comparisons with Remix. -** "The proposed method for DODA is exclusively used where augmentations are solely for one single class and do not encourage inter class interaction. Augmentation methods such as mixup and its improved version Remix (Chou et al. 2020) have proven to be efficient in data-imbalance settings. However, comparison of the proposed method with Remix + LTL methods is lacking."
>
> Thank you for your comments. Based on your suggestion, we have conducted comparative experiments on Remix, CUDA, and DODA on CIFAR-100-LT (IR = 50 and 100). The experimental results shown in the table below indicate:
>
> - All three augmentation strategies can achieve performance improvements under imbalanced data distribution, demonstrating that reasonable use of data augmentation can alleviate the long-tailed problem.
>
> - DODA achieved better performance than Remix, especially for tail classes, indicating that allowing each class to choose DAs beneficial to itself can avoid significant sacrifices.
>
> **Table: Accuracy (%) on CIFAR-100-LT dataset (IR = 50 and 100) wtih Remix.**
> |Method|Head|Medium|Tail|All (50)|Head|Medium|Tail|All (100)|
> |:-:|:-:|:-:|:-:|:-:|:-:|:-:|:-:|:-:|
> |CE + Remix|70.3|38.8|11.0|**46.7**|71.7|42.0|5.7 |**41.5**|
> |CE + CUDA |68.3|38.4|13.7|**46.2**|70.7|41.4|9.3 |**42.0**|
> |CE + DODA |71.2|40.3|12.6|**48.0**|74.8|43.8|10.0|**44.5**|
> |BS + Remix|62.8|47.7|41.2|**52.6**|64.0|50.7|27.4|**48.4**|
> |BS + CUDA |63.6|48.4|37.3|**52.7**|62.5|49.1|29.4|**47.9**|
> |BS + DODA |62.2|51.2|41.5|**54.0**|63.1|49.3|31.2|**48.7**|
>
> > **Comment 2: Optimality of the Selection of DAs. -** "The algorithm optimizes the augmentation strategy on 10 predefined augmentation functions with fixed strengths. The authors should discuss the optimality of the selection of these functions and provide ablation studies for the same."
>
> Thank you for your valuable comments. To verify the effectiveness of DODA, we have chosen ten widely used augmentation methods. Essentially, we could also choose more augmentation methods. In this case, the augmentation preference (or preference list) for each class may change, as they find DAs that are more beneficial to them. That is to say, when we add other augmentation methods, each class still tends to choose the most beneficial augmentation for itself.
>
> Based on your suggestion, on CIFAR-100-LT (IR = 100) and BS baseline, we have compared different candidate augmentation combinations, including 10, 15, and 20, and the corresponding experimental results are shown in the following table. We can find that DODA is still effective even though the predefined augmentation methods have changed. In addition, with the increase in the number of DAs, DODA may need more training overhead to obtain the optimal performance, i.e., it needs to iterate more on the weight matrix, and 200 epochs may not be enough to find the optimal weight.
>
> **Table: Accuracy (%) on CIFAR-100-LT dataset (IR = 100) wtih different candidate augmentation combinations.**
> |Method|Head|Medium|Tail|All (100)|
> |:-:|:-:|:-:|:-:|:-:|
> |BS|59.6|42.3|23.7|**42.8**|
> |BS + DODA w/ 10 DAs|63.1|49.3|31.2|**48.7**|
> |BS + DODA w/ 15 DAs|62.7|48.7|28.2|**47.9**|
> |BS + DODA w/ 20 DAs|63.1|50.0|28.6|**48.2**|
>
> We also provide specific augmentation methods.
>
>     l = [
>         (Flip, 0, 1),
>         (Mirror, 0, 1),
>         (EdgeEnhance, 0, 1),
>         (Detail, 0, 1),
>         (Smooth, 0, 1),
>         (AutoContrast, 0, 1),
>         (Equalize, 0, 1),
>         (Invert, 0, 1),
>         (GaussianBlur, 0, 2),
>         (Rotate, 0, 30), // 10
>
>         (ShearX, 0., 0.3),
>         (ShearY, 0., 0.3),
>         (Color, 0.1, 1.9),
>         (Brightness, 0.1, 1.9),
>         (Sharpness, 0.1, 1.9), // 15
>
>         (TranslateXabs, 0., 100),
>         (TranslateYabs, 0., 100),
>         (Posterize, 0, 4),
>         (Solarize, 0, 256),
>         (SolarizeAdd, 0, 110), //20
>         ]

---

> ### Author Response · Authors · 2023-11-19
> **Reply to Reviewer rRZN (II)**
>
> > **Comment 3: Further Comparisons with More Baselines. -** "The effectiveness of DODA on state-of-the-art long-tailed methods PaCo, and NCL is not demonstrated in the paper. It would be great if the results are provided in the rebuttal."
>
> Thanks so much for bringing these recent works to us.  Following your suggestions, we have analyzed these two papers to determine the feasibility of combining them with DODA. Due to computational resource constraints, we have replicated PoCa and NCL and conducted tests under two imbalance settings on CIFAR-100-LT. We respectfully argue two points:
>
> - For PoCa, based on the current experimental environment, we regret that we cannot directly run the official code, so we cannot provide quantitative results. However, we further analyzed the combinability of PoCa and DODA, and we found that through automatic augmentation of a single contrastive view, the two methods can achieve a good combination, and we continue to explore the subsequent results.
>
> - For NCL, we conducted comparative experiments on CIFAR-100-LT to verify the superiority of DODA, and the experimental results are shown in the following table. We can find that DODA has achieved better performance under the two settings, indicating that DODA can effectively alleviate this intertwined imbalance problem.
>
>     **Table: Accuracy (%) on CIFAR-100-LT dataset (IR = 50 and 100) wtih NCL.**
>     |Method|Head|Medium|Tail|All (50)|Head|Medium|Tail|All (100)|
>     |:-:|:-:|:-:|:-:|:-:|:-:|:-:|:-:|:-:|
>     |NCL + CUDA|68.9|52.5|47.8|**58.4**|69.2|55.2|34.4|**53.9**|
>     |NCL + DODA|68.1|53.2|49.2|**58.6**|69.4|55.6|35.7|**54.4**|
>
> > **Comment 4: Combineability in Contrastive Methods. -** "For contrastive methods such as BCL, how is DODA being used to generate multiple views of the same image for different classes?"
>
> Thank you for your comments. For contrastive learning-based methods, e.g., BCL, we define different views of the current input in the *getitem* function when we obtain the dataset. One view is obtained by sampling DA according to the augmentation weight distribution of each class in DODA, and the other is an augmented view obtained traditionally for all classes, i.e., according to the code provided by each contrastive learning baselines, a consistent augmentation is adopted.

---

> ### Comment · Reviewer_rRZN · 2023-11-22
> **Response to Authors Rebuttal**
>
> Dear Authors,
> As the paper received divergent reviews from other reviewers, I reviewed the response and paper again. I have the following concerns which are unaddressed:
>
> 1. PaCo code, which is claimed to be not reproducible, differs from my experience. The PaCo repository provides the exact commands that are needed to produce the result. Further, as the PaCo method is ICCV 2021, and the results produced by DODA are currently lower than that, it is hard to justify the SotA claim.
>
> 2. Unfair  Baseline Results. I thank you for providing results on NCL. However, the CIFAR-100 results recorded are only 0.1 (imb = 100) and 0.4 (imb = 50) more than the results reported by the NCL paper, I am still determining the significance of the improvement claimed here.
>
> Furthermore, there are a few other general concerns that I found while re-read the paper:
>
> 1. Sacrifice Rate Ambiguous: The definition of SR needs to be clarified. It seemed it determined with the percentage of classes where the model performance is lower than that of the baseline when the proposed DODA (/CUDA) method is combined with the baseline. However, in this case, even a small amount of lower accuracy (0.0001) can also lead to the counting of class as sacrificed. However, that is not true if we consider the statistical significance. Hence, the Sacrifice Rate should be determined in terms of the absolute accuracy difference. Furthermore, CUDA performance in absolute terms is similar to DODA; hence, it’s hard for me to see the difference.
>
> 2. Fair Comparison with CUDA: The authors propose that the role of augmentation should be studied in the literature. However, the CUDA paper studies it extensively. Hence, it would be good to have a section that lists reasons why DODA is better.
>
> 3. No experimental justification of theory in Sec. 2.2: The theory provided in Sec. 2.2 seems to be disjoint for the paper, as no experimental results are provided to support the theory. Hence, verifying its validity and significance is hard for me.
>
> Due to the above concerns, I will maintain my rating of the paper in its current form. I hope authors will use the feedback to improve their submissions.

---

> > ### Author Response · Authors · 2023-11-23
> > **Further Response to Reviewer rRZN**
> >
> > We appreciate and acknowledge your constructive comments. Considering that the rebuttal period is about to end, we here provide a brief explanation:
> >
> > - Indeed, PaCo provides the official code and corresponding commands. However, due to the fact that our current experimental environment does not seem to adapt well to the parallel computing of this code, we regret that we cannot obtain the final results in time. However, we are always looking for problems and trying our best to complete further experiments to demonstrate the effectiveness and flexibility of DODA. Although the rebuttal period is about to end, we will also display the subsequent results in the appendix.
> >
> > - To achieve the integration of DODA and NCL, we have reproduced and improved NCL. We found that the average accuracy of evaluated NCL (around 58.1 **[tail-47.3]** / 53.8 **[tail-33.5]**) has a slight decline compared to the provided results, so the improvements achieved by CUDA and DODA are essentially acceptable, especially in the tail classes (58.4 **[tail-47.8]** / 53.9 **[tail-34.3]** & 58.6 **[tail-49.2]** / 54.4 **[tail-35.7]**). In addition, *whether for PaCo or NCL, we hope that our method can alleviate the negative impact brought about by data augmentation, especially to avoid the hypocritical average performance improvement brought about by unfair augmentation and discrimination against disadvantaged classes*. It is more like a partner of the long-tailed learning method that supports inter-class fairness, providing help and improving the current method.
> >
> > - Thank you very much for your suggestion on the sacrifice rate. To evaluate the impact of different augmentation methods on different classes, we introduce the sacrifice rate as an intuitive evaluation indicator, which mainly counts the changes in the per-class accuracy before and after using the augmentation strategy. Of course, adopting the absolute accuracy difference may be a better choice. For the augmentation strategies being evaluated, these measurements are fair. By observing the accuracy changes caused by the use of augmentations and the relative changes between different methods, we can evaluate the inter-class fairness of different augmentation strategies, and find out whether some strategies achieve absolute accuracy improvement by pleasing other classes at the expense of some classes.
> >
> > - We are very grateful for your valuable insights. Discussing the role of augmentation in analyzing the advantages of DODA can make people understand the selection mechanism of DODA more intuitively, which complements the current analysis of 'sacrifice' and 'tendency', and better demonstrates the superiority of DODA. We will add further explanations to the existing analysis based on your suggestions. In addition, to verify the theoretical analysis, we have statistically more detailed changes in accuracy, e.g., Table 7 in the appendix. We found that the existing augmentations support long-tailed learning by bullying the weak, and the tail classes are more likely to become weak, which also supports our analysis in Section 2.2.
> >
> > Thanks again for your time, we will refer to your suggestions and add subsequent explanations and experimental results to improve our work.
> >
> > Many thanks!

---

### Official Review · Reviewer_PqHc · 2023-11-01

**Soundness:** 4 excellent
**Presentation:** 3 good
**Contribution:** 4 excellent
**Rating:** 8
**Confidence:** 5

**Summary:**

This paper makes a theoretical analysis of data augmentation strategies in long-tailed learning and points out that there may be a variety of potential interweaving imbalances in long-tailed learning, such as class and augmentation. The author further proposes a dynamic optional augmentation strategy to alleviate the above imbalance problem from a new perspective. The proposed method is tested on different main-stream long-tailed datasets, including dealing with imbalance problems and augmentation problems. The comparison with existing methods shows the superiority of the proposed method.

**Strengths:**

1. The motivation proposed in this paper on how to choose suitable data augmentation for different classes is practical, which innovates the traditional way of long-tailed learning.
2. The theoretical analysis is sufficient, and the author has analyzed the potential risks of data augmentation in long-tailed learning. The explanation that data augmentation may bring hypocritical performance improvements is interesting.
3. This paper conducted a large number of comparative experiments with methods of different tendencies on mainstream benchmarks, including analysis at different levels. The results can demonstrate the effectiveness of this method.
4. This paper is well written, and the dynamically optional method proposed is easy to understand. In particular, the visualization of the training process allows me to have a more intuitive understanding.

**Weaknesses:**

1. My main question or weakness is that although the author has conducted analysis and verification in the traditional long-tailed learning field, the long-tailed distribution may not only exist in the visual field, and some other fields (e.g., graph cls.) seem to have the same problem. This paper lacks a discussion on the scalability of the proposed method.
2. Section 4.3 mentions that 'DODA can avoid more classes being sacrificed', but it seems that this cannot be intuitively found based on Figure 4.

**Questions:**

1. It would be beneficial if the author could provide a brief explanation or discussion on the scalability of the proposed method, which could promote its application in other open scenarios.
2. Regarding Section 4.3, the authors mention that DODA can avoid more classes being sacrificed. However, it may not be intuitive to find this based on Figure 4. Therefore, it would be helpful if the author could provide a more detailed explanation of the model's effect, which would allow others to better understand the improvements brought about by this method.

---

> ### Author Response · Authors · 2023-11-19
> **Reply to Reviewer PqHc**
>
> We appreciate your comments. To address your concerns, below we provide the point-to-point responses to carefully illustrate the scalability of our method and provide additional explanations. We will carefully revise our paper by taking into account all your suggestions. Looking forward to discussing more with you.
>
> > **Comment 1: Scalability of DODA —** "It would be beneficial if the author could provide a brief explanation or discussion on the scalability of the proposed method, which could promote its application in other open scenarios."
>
> Thanks for your comments. Indeed, the long-tailed distribution exists in various domains, not limited to the vision domain. We strongly agree that we need to analyze the scalability of DODA. Data augmentation is a common technique in different domains, e.g., graph learning and NLP, and there is a variety of DAs in these domains, although perhaps less abundant than in vision. Therefore, when faced with an imbalance challenge, using DODA to select the appropriate enhancement for each class can also help mitigate this challenge in these domains. In addition, the tasks and datasets that are well-established or mainstream in long-tailed learning are often confined to the vision domain, and we also try to establish standard datasets in other domains to verify the effectiveness of DODA. Considering the differences in data types and the specificity of data augmentation methods across different domains, it is unknown whether the performance of DODA would fluctuate, which motivates us to continue exploring.
>
>
> > **Comment 2: Intuitive Explanation of Figure 4 —** "Regarding Section 4.3, the authors mention that DODA can avoid more classes being sacrificed. However, it may not be intuitive to find this based on Figure 4. Therefore, it would be helpful if the author could provide a more detailed explanation of the model's effect, which would allow others to better understand the improvements brought about by this method."
>
> Sorry for the confusion. By observing Figure 2 and Figure 4 simultaneously, we can observe that the sacrifice rate of DODA is much lower, indicating that it can significantly alleviate the sacrifice problem. We also display the sacrifice rates of different methods in the following table so that you can more intuitively confirm the effectiveness of DODA.
>
> **Table: Sacrifice rate (%) under diffirent DAs on CAFIR-100-LT dataset. $\delta$ is the relative improvement.**
> |IR   |CE + Cutout|CE + CUDA|CE + CUDA &Cutout|CE + DODA | $\delta$ v.s. CUDA
> |:-:|:-:|:-:|:-:|:-:|:-:|
> |$50$    |$31\\%$|$38\\%$|$27\\%$ |$7\\%$|$-31\\% p$|
> |$100$ |$21\\%$|$29\\%$|$23\\%$ |$5\\%$ |$-24\\% p$|

---

### Author Response · Authors · 2023-11-19
**Author Rebuttal**

We appreciate all the reviewers for their valuable comments and suggestions. We are encouraged that they find our motivation practical and clear (Reviewer PqHc, ZxxQ), our analysis insightful (Reviewer PqHc, rn58), our method effective (Reviewer rRZN, rn58), our experiments extensive (Reviewer PqHc, ZxxQ, rn58), and our representation clear and well-organized (Reviewer PqHc, rRZN, rn58). This helped improve our submission and strengthen our claims. We have tried our best to address the main concerns raised by reviewers and we hope that these improvements will be taken into consideration. We also present the point-to-point responses for each reviewer below.

---

### Meta-Review · Area_Chair_kWzN · 2023-12-06

**Metareview:**

This paper received an overall accept average with two accepts (8, 8) and two marginal scores (5, 6).  The reviewers overall found the paper well written, easy to follow and well-motivated.  Long-tailed data is common in practice, and thus identifying and correcting issues with data-augmentation in the long-tailed setting seems valuable and relevant to the ICLR community.  Multiple reviewers raised concerns regarding missing baselines, and some simple ablations.  However, otherwise the reviewers found the empirical evaluation convincing.  The experiments on long-tailed versions of CIFAR-100 and Imagenet were interesting, and the proposed method seems to consistently improve over the baselines.  In discussion, one reviewer raised concerns regarding the computation of the sacrifice rate (SR).  That reviewer seemed to believe that it would be better to evaluate as e.g., CE + CUDA and CE + DODA, rather than evaluating their %p difference in SR values based on CE.  Another reviewer felt that comparison would be insightful as well.  Given two very positive reviews and the absence of significant concerns against acceptance would suggest the paper could be exciting to a subset of the community.  Therefore, the recommendation is to accept.  Hopefully the reviews will help improve the paper for the camera-ready version.

**Justification For Why Not Higher Score:**

The reservations of two of the reviewers, particularly regarding the presented metrics in the experiments, suggests to me that the paper isn't quite up to the level of a spotlight or oral.

**Justification For Why Not Lower Score:**

Given two very positive reviews and the absence of significant concerns against acceptance would suggest the paper could be exciting to a subset of the community.

---

### Decision · Program_Chairs · 2024-01-16

Accept (poster)